# Sexual Violence against Adults Aged 50 Years and Older and Implications for Prevention: A Thematic Analysis of Service Providers’ Perceptions

**DOI:** 10.3390/ijerph21091220

**Published:** 2024-09-17

**Authors:** Michelle D. Hand, Mo Yee Lee, Michelle L. Kaiser, Cecilia Mengo, Holly Dabelko-Schoeny

**Affiliations:** 1Department of Social Work, George Mason University, Fairfax, VA 22030, USA; 2College of Social Work, The Ohio State University, Columbus, OH 43210, USA

**Keywords:** abuse/neglect, gender, sexual violence, qualitative analysis, theory, women’s issues

## Abstract

At-risk older adults and older survivors of sexual violence (SV) remain largely absent from SV prevention and intervention, owing to ageism and sexism, as well as other intersectional forms of prejudice, including among service providers (e.g., social workers, healthcare professionals, practitioners in SV organizations, and practitioners who serve older adults). This study explored perceptions, knowledge and experiences with SV against adults 50 years and older. Service providers who work with older adults and/or survivors were recruited, owing to where SV in later life is reported (e.g., healthcare, long-term care, and social service organizations, and to police in addition to SV service organizations), to contribute to the limited research in this area and to advance prevention and intervention. A survey was conducted on SV in later life, exploring knowledge, perceptions and experiences with SV in later life along with potential solutions for prevention and intervention among 126 service providers who worked with survivors and/or older adults. Their responses were thematically analyzed. Five themes were identified: (a) misconceptions of SV in later life and unique barriers to preventing it; (b) needs for knowledge, awareness, research and education; (c) policy and resource development; (d) victim blame and internalized stigma, and (e) ageism, intersectional prejudice and rape culture. The findings offer an in-depth understanding of barriers to prevention, and intervention, and multi-level recommendations for addressing them, which are provided by a diverse group of service providers who have worked with older adults and/or with survivors, reflecting multidisciplinary practice wisdom and experience.

## 1. Introduction

While sexual violence (SV) occurs past 50 years of age, older adults who are at risk of SV and older survivors of SV have been largely excluded from research, policy, practice and education surrounding prevention and intervention, as adults aged 50 years and older are not perceived as sexual or desirable by society in comparison to their younger counterparts, owing to ageism [1,2,3,4,5]. All older adults remain at risk for SV, and the same factors that place adults at greater risk for SV earlier in life, largely owing to influences of prejudice and discrimination on violence, can intersect with ageism and other forms of prejudice to raise risks for SV in later life [1,2,3]. For example, sexism remains a chief risk factor with women and transgender individuals being at a greater risk for SV across the life course [1,2]. Experiencing SV earlier also raises risk for SV across the life course [2]. Further, disabilities, like dementia, not only raise risks for SV but can also serve as a barrier to its prevention, in consideration of ableist beliefs that people with dementia cannot be victimized [1,2,5], which will be further discussed.

### 1.1. Defining Sexual Violence (SV)

Although a widely agreed upon definition of SV is still in development [1], SV is generally understood across the life course as an overarching term that captures a range of conduct, which may involve non-physical behaviors, like unwanted sexual remarks, or hands-off SV, to physical behavior, such as rape, or hands-on SV [6]. SV remains the least acknowledged, discovered or reported form of elder abuse [7,8,9]. In describing individuals who have survived SV, “victims” and “survivors” will be used interchangeably to respect the different ways people prefer to identify.

### 1.2. Defining Later Life

The age at which later life begins, commonly identified as 60 years or older, is important to consider when studying SV in later life, which is often referred to as elder sexual abuse (defined as an unwanted hands-on or hands-off sexual interaction with an older adult) [10]. Gerontologists do not agree on when later life begins or, in turn, when to begin studying SV in later life; thus, the cut-off age in research on SV in later life has varied, as some research has explored the experiences of individuals who are as young as 50 years, owing to findings that suggest non-dominant populations may age faster due to structural discrimination and related trauma [11,12,13]. This is relevant to research on SV in later life, as SV and resulting trauma (a natural yet complex response to an event or events that overwhelm one’s sense of equilibrium due to perceived or actual limited social supports, often through posing threats of serious injury, death, and/or violation of physical integrity) have also been linked with poverty, discrimination, poor health, obesity, and accelerated aging [14]. In other words, aging can depend on various personal experiences (e.g., structural oppression, poverty, and other factors linked with risks for violence) as well as earlier life SV exposure and the physical and mental health conditions that survivors often experience (e.g., poor general health, depression, low-self-esteem, eating disorders, obesity, and obesity-related disorders, like diabetes) [11,12,13,14]. Accordingly, it has been suggested that studying SV among adults who are 50 and older is likely to be more inclusive of underserved populations because some people may age faster than others [12].

As such, this manuscript will focus on adults who are 50+ as the age cut-off for defining SV in later life, in adopting the inclusive later life definition provided by The National Clearinghouse on Abuse in Later Life (NCALL), for the reasons discussed above [12]. Further, the focus of this manuscript is on SV in later life in particular, in consideration of it being widely underrecognized, underreported, and understudied, despite SV in later life being a longstanding, growing public health issue [4] with lasting physical and psychological consequences, which service providers (e.g., healthcare and long-term care workers and ombudsmen, social workers, SV service providers, and law enforcement) can help to prevent and address [5,6,8,9]. Service providers can help achieve this by preventing SV from occurring in later life, which can in turn prevent the sexual trauma and lasting consequences that result from it, and service providers can also treat the lasting consequences of SV in later life [2]. The timely relevance of SV in later life to service providers, both those who serve older adults, and those who serve SV survivors, will be further described within this section.

### 1.3. Prevalence and Significance

A need for reliable prevalence estimates of SV among older adults in community-based or institutional settings remains, as research in this area is still emerging [1,2,7,13]. Still, SV among older adults has been documented for over 30 years [11]. Population studies have resulted in the lowest prevalence estimates with a range of 0.2 to 5.2% [1]. Yet, studies across age groups have yielded the highest prevalence rates, ranging up to 17% of all SV cases [1]. Recently, Nobels and colleagues (2021) estimated a 44% lifetime prevalence of SV among adults aged 70 years and older, noting an 8.4% reported prevalence within 12 months of their study [15]. Still, the variation and range discrepancies across available studies demonstrate variations in sample sizes and reporting by discipline, along with remaining needs for research [1,2,7]. SV is underreported across age groups; as such, while research is still emerging, researchers across disciplines have highlighted that the prevalence of SV in later life is likely underestimated [1,2,15].

This has substantial implications for service providers who serve older adults and for service providers who meet the needs of victims of SV [1,2,5,16]. Service providers can offer valuable information surrounding barriers to prevention and intervention as well as on what is needed to address them [1,2,5]. However, very little research exists on the experiences of service providers who respond to SV in later life or on how their practice wisdom may be used to help advance prevention and intervention [1,2]. Service providers can help enhance our understanding of SV as researchers, which can result in improved policies to prevent and address SV in later life [2,6]. A more accurate understanding of SV in later life could be used to inform research on prevalence, and service providers themselves can help provide more precise prevalence estimates in consideration of roughly 30% of the SV cases in later life remaining unreported and service providers learning of unreported SV [2,11]. Additionally, a more accurate understanding of SV in later life (e.g., gained through including service providers from multiple disciplines in research on SV in later life) could influence practice and the trainings practitioners receive [2]. A more well-rounded understanding of SV in later life could also be used to work toward a transdisciplinary understanding of SV in later life, based on service provider knowledge and wisdom [2,6].

In terms of where SV in later life is reported, the limited existing research in this area has demonstrated that reports of SV in later life have been made to service providers working in multiple disciplines [2]. This includes healthcare workers [8], adult protective services (APS) workers and ombudsmen [9], practitioners in long-term care settings [5], social workers [16], service providers in organizations that service SV survivors (e.g., rape crisis centers) [17], and police [16,18]. The WHO (2016) has referred to each of these diverse service providers (healthcare providers, social service workers, law enforcement, and victim service workers), many of whom regularly witness SV and of whom receive reports of SV, as “key stakeholders”, as these service providers are essential to its prevention and intervention, [6]. Across the lifespan, and in especially in later life, further multidisciplinary focus is also needed to prevent and address SV in later life [1,2,16].

Societal perceptions have long been acknowledged as contributing risk factors for elder abuse [19]. As the WHO (2002) noted, “cultural norms and traditions… such as ageism, sexism and a culture of violence… are also… recognized as playing an important underlying role” [20] (p. 132). For example, Brozowski and Hall (2010) have pointed out that at the structural level, “global aging is juxtaposed with neoconservative global market forces which encroach on many Western nation welfare policies designed to protect citizens, including social security, healthcare, and social services for older adults”, who have been referred to “as an expensive burden to the state” [16], p. 1184. This sentiment was particularly prominent during the COVID–19 pandemic, during which time older adults, especially those who lived on the margins owing to structural oppression, or widespread discrimination, disproportionately lost their lives as a result of ageist beliefs and public health priorities [3]. Such forms of structural discrimination and oppression (referred to as structural ageism when older adults are disproportionately impacted by preventable social problems) can influence the development or lack of resources and human rights (e.g., the right to health and to safety from violence, including from SV in later life) [3,16,18]. Moreover, older adults are often publicly perceived as incompetent and undesirable, and ageist portrayals along with youth-focused rape stereotypes may explain why older adults are generally excluded from efforts to prevent and address SV [1,2,13].

Specifically relating to the structural impacts of public perceptions, older adults are often publicly perceived as incompetent and undesirable; ageist portrayals and youth-focused rape stereotypes may explain why older adults are generally excluded from efforts to prevent and address SV [1,2,13]. Beyond impacting policy development, public perceptions of SV in later life may also become internalized by survivors [21]. Social acceptance of SV as a private issue can impede intervention while encouraging silence, stigma, and posttraumatic stress, along with limited reporting when SV in later life occurs [2,3,21].

At the organizational level, research has demonstrated that healthcare providers regularly dismiss the concerns of older women [22,23]. Older women have reported stigma when accessing medical services, owing to racism, sexism, heterosexism, ableism, classism, and/or other forms of prejudice among service providers from multiple disciplines [22]. This is problematic, as violence and related health concerns are disproportionately experienced by women of color across the lifespan and in later life [22].

To add to this, service providers in SV organizations have expressed shock in response to SV in later life, owing to needs for further awareness that older adults can be victimized [1]. Research has demonstrated that service providers in nursing homes have been similarly surprised by the possibility that older adults can be considered sexually desirable and have not considered SV to be a problem in later life [5]. Pressing needs have been established for further knowledge, awareness, and training for service providers on SV in later life, along with a need for further research with service providers to advance prevention and intervention [1,5]. Service providers, particularly those working in healthcare, SV, and long-term care organizations, are especially well positioned to aid in the prevention and intervention of SV toward older victims [1,3,5,24].

In consideration of this and of older adults being largely excluded from SV prevention and intervention, further research is urgently needed to address widely accepted beliefs that older adults are seldom or altogether not sexually abused [5]. To address this and the taboo nature of SV past 50 years of age [1,2,5], a survey involving open-ended, open-text questions was conducted to gain a qualitative in-depth understanding of perceptions of SV past 50 years of age, age- and gender-based power dynamics, and how they may relate to potential barriers to and solutions for prevention. Data were collected from healthcare and social service providers who regularly serve older adults and SV survivors.

The decision to focus on the knowledge, perceptions, and experiences of practitioners who serve older adults and those who serve victims was made owing to the ability of the various practitioners who receive reports of SV in later life (e.g., healthcare workers, long-term care workers, social workers, SV service providers, police, and ombudsmen [5,8,9,16,17,18]) to hinder or advance prevention and intervention regardless of discipline. Moreover, as discussed, limited research on SV in later life exists with practitioners, and the research that does exist suggests that many practitioners from multiple disciplines are not aware that SV is a problem in later life, and they are often unaware of how to prevent or address it when it is reported to them [1,2,5]. Learning about any potential awareness needs and challenges with preventing and addressing SV could offer invaluable insight for what is needed to prevent and address the growing yet hidden and under-researched global health problem of SV in later life.

### 1.4. Present Study

This research, which will be further discussed in the next section, was conducted as part of a larger mixed-methods dissertation study, which aimed to explore societal perceptions of and personal experiences with SV and, within this aim, to explore knowledge, perceptions and experiences of SV in later life as well as age- and gender-based power dynamics and how they may relate to barriers to and solutions for preventing SV in later life. The 27 survey questions, regarding service provider knowledge, perceptions, and personal experiences with SV, were asked to gather in-depth information on potential influences on how service providers understand and respond to SV in later life. For example, considering that one in three women experience SV [21] and many helping professionals are women, it is possible that some practitioners may have experienced SV, which could impact how they understand and respond to SV in later life (positively, e.g., through believing victims, or negatively, e.g., owing to internalized victim blame, which is common among SV survivors [1,2]).

The primary purpose of the study is to explore knowledge, experiences and perspectives of SV past 50 years of age among individuals who have directly worked with adults in this age group and/or with SV survivors. This study also aimed to explore how power dynamics linked with age and gender may impact perceptions on SV past 50 years of age among service providers, possible barriers to prevention, and how any such barriers may be addressed, as a timely contribution to the limited available research in this area.

Service providers are essential to prevention and intervention; for example, their perceptions and knowledge on SV in later life and any needs for further awareness in this area among these professionals can impact service delivery [1,2,5]. However, the authors recognize that survivors themselves are key experts on SV in later life. In consideration of this, data were also gathered from survivors and disseminated as part of the first author’s larger dissertation study. In the present manuscript, which is based on the first author’s dissertation study, the findings are presented on the data collected from service providers, who have worked directly with older adults and/or with SV survivors (including healthcare and long-term care workers and ombudsmen, social workers, SV service providers, and law enforcement) owing to these service providers being defined as key stakeholders in preventing and addressing SV [6], including in later life [5,8,9,16,17,18].

It is noteworthy that not all service providers that were included in this study exclusively served SV survivors, because research has demonstrated that SV in later life is directly witnessed by healthcare professionals and social service professionals, and it is reported to these professionals as well as to SV organization workers, law enforcement, long-term care workers and administrators, and ombudsmen [5,8,9,16,17,18]. Across the lifespan, and in especially in later life, needs for further multidisciplinary focus have also been identified to more comprehensively prevent and address SV in later life [1,2,16]. However, as discussed, there is little published research on service provider awareness, knowledge and experiences with learning about SV in later life, and the present study contributed to the limited knowledgebase in this area [1,2].

To achieve the study aims of offering an in-depth understanding of service providers’ knowledge and perceptions of SV in later life, and what is needed to prevent it, the following research questions were explored:What are the knowledge, experiences and perceptions of SV in later life among individuals who have directly worked with older clients or victims of SV?How (if at all) may power dynamics linked with social factors such as age and gender impact knowledge and perceptions on SV in later life, according to service providers who have directly served older clients and/or victims of SV?What (if any) barriers may exist to preventing and addressing SV in later life, as identified by service providers who have worked with older and/or with victims of SV?If barriers to prevention and intervention exist, how may they be addressed, as recommended by service providers who have worked with older adults and/or with victims of SV?

## 2. Materials and Methods

### 2.1. Conceptual Framework

A Critical Feminist Gerontological–Social Ecological Framework, comprising the Critical Feminist Gerontological Framework and the Social Ecological Model (SEM) [2], informed the survey questions and guided the research process. This integrative Critical Feminist Gerontological–Social Ecological Framework underscores multi-level and intersectional discrimination of older adults, women and members of other non-dominant populations owing to structural oppression [1,2,10]. Accordingly, this study explored power dynamics, between men and women, older and younger adults, and dynamics that are dependent on culture and intersectional discrimination surrounding SV past 50 years of age [1]. A key aim was also to empower service providers who work with survivors past 50 years of age by encouraging them to share their experiences, knowledge and perspectives on what (if anything) may be needed to influence meaningful changes relating to how SV is approached past 50 years of age.

The SEM encourages the exploration of links between individuals, relationships, communities, organizations and society along with multi-level impacts on wellness. This approach informed the survey questions while allowing for potential to gather data on inter-relational, communal, and organizational-level responses in following recommendations for incorporating the SEM [10]. Still, the main focus of this study was on individual experiences with or knowledge of SV past 50 years of age, related resources, and the potential impacts of relational, communal, or societal responses. These service providers were recruited because of their individual knowledge and experiences as well as those encountered in social service and healthcare organizations. The importance of exploring the best strategies for addressing ageism among practitioners from multiple disciplines has been well documented, owing to ageism and intersectional prejudice toward older adults impacting prevention and intervention across social service and health disciplines [1,3,7].

### 2.2. Participants and Procedure

Data were collected through a survey via Amazon MTURK. Participants were directed to a Qualtrics link and asked up to 27 open-text questions to garner qualitative data on their response to a randomized SV vignette, prevention needs, beliefs related to SV, resource knowledge, and experiences with SV past 50 years. The vignettes described SV scenarios involving victims up to 81 years old, as part of a larger dissertation study, to gauge how a victim’s age may impact perceptions of SV (which all participants in the larger dissertation study were asked prior to being asked if they worked with older adults or with survivors and clarifying their occupation). The larger mixed-methods dissertation study involved quantitative questions regarding responses to the SV vignettes to examine how the age of the victim and the type of SV may potentially impact the perceived seriousness, culpability, reportability and knowledge of SV. Following this, a thematic analysis was conducted of the open-text (write-in) survey responses of survivors of later life SV and of service providers who participated in a larger dissertation survey. Service providers who did not personally experience SV were asked 24 of the 27 survey questions, and if they had personally experienced SV, they were asked an additional four questions to explore how (if at all) personal SV experience may have impacted how they understood and responded to SV as part of the larger dissertation study (see Appendix A for the survey questions). For the larger dissertation study that this manuscript is based upon, questions were asked both generally about SV and about SV against people who are 50+ years of age. For the purposes of this manuscript, the analysis of the latter questions, focused on SV against people who are 50+ years of age, are reported.

Participants were drawn from a convenience sample of MTURK workers in consideration of stigma linked with SV among elders and to minimize potential harm in a low-risk confidential space for offering information on this complex social taboo. The sample size was based not only on the qualitative data and analysis that would be needed to answer the qualitative research questions as part of the larger dissertation study but also on the minimum number of participants that would be needed to conduct the quantitative analyses for the dissertation study.

Thus, a statistical approach was used to determine the sample size based on the appropriate number of participants that would to be sampled in order to conduct each quantitative analysis. This was 500 people for the quantitative analysis with the highest range for qualitative research reaching as many as 400 or more participants. The 500 participants we aimed to recruit for the quantitative analyses aligned with our qualitative aims as well, as in order to conduct a thematic analysis, 10–50 people are recommended for text that is provided by participants (e.g., through open text survey questions, such as in the survey) [24]. Multiplied by the five kinds of work industry for service providers (healthcare, social work, work in an SV organization, long-term care, or law enforcement), this equaled between 50 and 500 participants for collecting the needed data to establish common patterns that could answer our research questions while not collecting too much data to manage [24].

The service providers whose open-text responses were qualitatively analyzed were a subsample of individuals who participated in the larger mixed-methods dissertation survey, who indicated that they were healthcare workers, social workers, long-term care workers, SV agency workers, ombudsmen and law enforcement officers. Further details on the sample are provided within the Results section.

### 2.3. Inclusion and Exclusion Criteria

Included were participants who selected that they worked with older adults and/or with SV survivors in response to demographic questions on work type and individuals who indicated they were healthcare workers, social workers, long-term care workers, administrators and ombudsmen, SV agency workers, and law enforcement officers, who are likely to have worked with older adults and SV survivors, owing to the multidisciplinary needs for information on SV past 50 years established in research, and the likelihood of encountering it, considering the nature of SV past 50 years and how reports are made [2].

Thus, the sample included healthcare and social service workers who have worked with older adults and/or with survivors of SV. Like the other service providers, law enforcement receive SV reports [18], collaboration with them is recommended for prevention, and law enforcement have been identified with other service providers as “key stakeholders” for preventing and addressing SV [6]. While it is possible that law enforcement may have different views from healthcare providers on SV in later life, aside from the professional language used “e.g., ‘assailant’ instead of ‘perpetrator’ for example”, there were no substantial differences found among the different professions with regard to the nature of their responses on their views of SV in later life or what is needed for prevention and intervention, including among the six law enforcement officers included in the study. This was determined by closely reviewing the responses of participants from each profession, organized by profession in Microsoft Excel, to explore potential differences between responses. No substantial differences were found regarding how they defined SV in later life, barriers to prevention, or what they believed was needed to prevent it. As such, the decision was made to include all of the different service providers’ responses in the thematic analysis.

Excluded were individuals who answered less than half of the survey questions, which are listed in Appendix A.

### 2.4. Online Qualitative Research

Research has begun to demonstrate the benefits of online qualitative data collection to explore topics that may be difficult to discuss in person, to accommodate multiple schedules, and to eliminate transportation barriers [25,26]. Online survey data can yield in-depth insights that may not be possible during interviews when studying issues that may be complex or taboo [26]. Thus, researchers are beginning to gather qualitative data online, through MTURK, to explore sensitive topics, such as attitudes toward SV [25], which can be less intimidating than exploring SV during in-person focus group or interviews. In this study, open-text questions within a survey used for a larger dissertation study were thematically analyzed. The larger dissertation study was a mixed-methods study which involved large-scale quantitative analysis, and open-text questions were used to garner qualitative data for thematic analysis of service providers’ knowledge, perceptions and experiences with SV in later life. The themes from this qualitative analysis are captured in this manuscript.

While this strategy of collecting qualitative data through open-text survey questions has been useful for studying violence against vulnerable populations owing to the sensitive nature of this topic [24], it does limit the questions that could be asked to structured questions. Still, most of the questions were open in nature, and participants were asked what else the researchers should know that they had not been asked. All the questions in the survey were phrased to be easily understood (e.g., using ‘unwanted sexual touch’ for example, rather than ‘sexual violence’, with the term ‘unwanted sexual touch’ having been used by other researchers as well to briefly describe SV in plain language [27]). The survey included general questions about SV (e.g., ‘What (if anything) may people need to know about unwanted sexual touch or unwanted sexual behavior?’) as well as very specific questions about SV against people ages 50 and older (e.g., ‘What (if anything) may be needed to prevent unwanted sexual touch or behavior for people ages 50 and older?’). Still, it is worth mentioning that some questions cannot qualitatively assess people’s understanding and experience with SV against adults who are older than 50, as some of the questions pertained to other parts of the larger dissertation study that are not discussed in this manuscript [28].

Only those questions that related specifically to people’s views on survivors aged 50 or over were included in the analysis that is reported in this manuscript. A list of the survey questions is provided in Appendix A, with an asterisk to the left of those questions pertaining to the analysis that is reported in this manuscript, focused on service provider knowledge on SV in later life, barriers to prevention, and recommendations for preventing SV against adults older than 50 years.

### 2.5. Ethical Issues

Approval for this study was granted by the Institutional Review Board (IRB) at The Ohio State University. A trigger warning was also provided along with the contact information for the research team and select mental health resources.

### 2.6. Thematic Analysis

Responses were collected from surveys completed by participants who identified as service providers working in healthcare-related fields, social work, SV organizations, long-term care or law enforcement were thematically analyzed. Details regarding the participant sample are given in the Results section.

Thematic analysis is used to answer specific questions through analytical themes that are explored across a large data set [29]. Following the steps for thematic analysis outlined by Braun and Clark (2006), themes were identified by the first author as part of a dissertation study across the data set by reading and re-reading the words of participants to promote familiarization with the data, first in Microsoft Excel, and later, in NVivo [26]. Initial codes were identified using descriptive, in vivo and values codes; were revised; and then were combined to yield additional themes [29]. These themes were reviewed and revised to closely reflect participant language; then, they were combined and refined, resulting in higher-level themes that captured the participants’ words [29].

The first author conducted the initial coding, which was discussed and consulted with the research team. Credibility was ensured by gathering information from a variety of sources, which was inclusive of a multidisciplinary group of individuals who work with elders and/or SV survivors, such as social workers and healthcare workers [30]. Transferability, dependability and confirmability were increased through a detailed audit trail [31]. Trustworthiness was further increased by including direct quotes during coding and analysis [31].

## 3. Results

Responses from 126 participants were thematically analyzed. Of these, 70 were healthcare providers or social service workers who did not work in healthcare-related fields (e.g., public health workers), 20 were social workers, 21 were individuals who worked in SV organizations, nine were long-term care administrators or ombudsmen, and six worked in law enforcement. In this study, these workers are collectively referred to as service providers or participants.

Of these 126 service providers, 26 (21%) indicated that they worked once or more per week with SV survivors, 34 (27%) indicated that they worked once or more per month with SV survivors, 32 (25%) indicated that they worked once or more per year with survivors, and 34 (27%) indicated that they did not work with SV survivors at all in their field of work, although they did work in fields that served older adults (who could be victimized). So, the majority of the service providers (73%) indicated that they did work with SV survivors in some capacity as part of their work, although 27% were not answering the questions based on direct practice experience with survivors but rather in the context of working with older adults who may be at risk of SV. Of those 27% who indicated they did not directly work with survivors as part of their job, three noted personally experiencing SV since turning 50: one shared that they learned of a coworker being sexually abused after turning 50, one shared that they learned of a friend being sexually abused after turning 50, two had romantic partners who experienced SV after turning 50, and one indicated learning of someone they know being sexually abused after turning 50 but did not disclose their relationship to the victim. Thus, eight (nearly 30%) of these participants still either directly experienced SV after turning 50 or personally knew someone who did, which may demonstrate the personal relevance of the problem of SV in later life beyond the professional relevance of this public health issue, owing to their work with older adults. Highlighting personal relevance may also be useful for consideration when developing bystander trainings to inform service providers of the problem of SV in later life and why it is a significant problem that service providers should learn about even if they do not knowingly work with survivors.

The majority of the sample (*n* = 104) was younger than 50 years, while 22 were 50 years or older. Most were men (*n* = 69), 56 were women, and one service provider was transgender. Nearly half of the service providers were White (*n* = 60), while most were members of non-dominant racial or ethnic groups: 35 were Black or African American, 12 were Asian, eight were U.S. Indigenous or Native American, five were Latino or of Spanish origin, four were White African, one was Middle Eastern, and one indicated another racial background. While most participants identified as heterosexual (*n* = 97), 3 identified as gay or lesbian, 22 identified as bisexual, 1 identified as pansexual, 2 identified as asexual, and 1 preferred not to share their orientation. While most of the service providers were not SV survivors, 34 (26.98%) indicated that they were survivors of SV. Table A2 provides an overview of these demographic characteristics.

Five overarching themes were identified. These included misconceptions of SV in later life and unique barriers to preventing it; needs for knowledge, awareness, research and education; policy and resource development, victim blame and internalized stigma, and ageism, intersectional prejudice (based on various identities that are marginalized owing to structural oppression), and rape culture [32]. These themes will be discussed in the following sections.

### 3.1. Misconceptions of Sexual Violence in Later Life and Unique Barriers to Preventing It

There was an emphasis on SV being a complex and pervasive problem across the lifespan with some participants suggesting that the age of the victim is not relevant. As one participant put it, “Is it common for 50 and older? [It is] wrong for any age”. Still, other participants noted that they had not considered SV to be an issue for adults who were in their 50s and older prior to participating in this study, as one participant admitted, “I usually think of people my own age (younger) but [I] never thought of how older folks manage [SV], or if they had experienced [SV] when they were younger”. Similarly, a cited barrier to preventing SV against adults who are 50 and older was that SV against older adults is not commonly discussed in public. Additional barriers to preventing and addressing SV in later life are described in the following four sub-themes, surrounding (a) stereotypes of SV and older adults, (b) medical issues and credibility, (c) vulnerability, and (d) generational influences on what constitutes SV.

#### 3.1.1. Stereotypes of Sexual Violence and of Older Adults

The results suggest that misunderstandings about the nature of SV and who can be victimized have also been influenced by stereotypes about SV as well as about older adults and the aging process. For example, with regard to SV stereotypes, as one participant put it, “rape or child molestation are generally considered the two extremes of sexual abuse”, suggesting that further understanding on what constitutes SV is needed. Other participants similarly reflected this. As an example, similarly, a social worker noted that when thinking of SV overall, “the first thing that comes to mind is an adult abusing a child”. And when reflecting on prevention needs for SV in later life, another healthcare worker suggested that SV is “important to particularly children and woman”. This may reflect stereotypes about who is impacted by SV, leaving older adults out of the narrative, and in this case, out of future recommendations for prevention despite the focus of the study being on SV in later life.

It is noteworthy that when child sexual abuse was introduced in response to the questions that were analyzed, which were focused on SV in later life, child sexual abuse was most often referenced by SV agency workers, social workers, and healthcare workers, although one long-term care administrator suggested a need for SV prevention to begin in childhood (which was a common sentiment, but this was mostly among SV agency workers and social workers as well as among some healthcare providers). For example, SV agency workers noted early life SV can increase risks for SV later in life. And some healthcare workers underscored the importance of teaching children early on about how to avoid SV (e.g., through refusal to engage in sexual activities, placing responsibility at the individual level on children and parents and at the organizational level on schools and religious leaders to prevent risks for SV that can impact later life).

In addition to this, a healthcare provider recommended, “I think the best thing to do is to make sure people of all ages know how important it is to not allow strangers being in their house if they do not know them, especially if they are seniors and disadvantaged in any way”, both reflecting a belief that most SV is stranger-based and that frailty in later life is common.

#### 3.1.2. Medical Issues and Credibility

Dementia was noted as adding to the complexity of SV in later life, as for example, one healthcare provider shared that SV against people who are 50 years and older “may not be believed or written off due to medication effects or dementia”. Dementia has also been highlighted in research as a barrier to prevention in later life, owing to concerns surrounding credibility among people living with dementia which can raise risks for SV [1,2]. All service providers who referenced dementia as a barrier to preventing and addressing SV in later life worked in healthcare, which may suggest that healthcare workers are more aware of unique risk factors for SV and barriers to preventing and addressing it among older adults. Further, determining consent, including for medical examinations following SV, is a unique issue for people living with dementia [1].

#### 3.1.3. Vulnerability

While SV was considered a major problem across the lifespan, some participants believed that older adults were more vulnerable to SV, or as one participant put it, “easier to be taken advantage of”. Another participant added, “Older people are most likely attacked because they believe no one will believe them”, Other participants cited structural issues, with SV not being taken seriously, which will be described in a separate theme. Thus, the reasons for the greater perceived vulnerability to SV among adults who are 50 years and older varied among participants.

#### 3.1.4. Generational Influences on What Constitutes Sexual Violence

The participants also shared that understandings of SV can vary depending on culture, and it was noted that “older people grew up in a time that unwanted sexual behavior was looked at much differently than it is today”, highlighting potential generational differences in how SV is understood.

SV definitions also varied across the sample. Remaining needs for a clear and shared definition of SV have also been reflected by researchers [1,2]. Still, it is noteworthy that understandings of personal experiences with SV and sexual harassment and perceptions of SV against older clients can be influenced by intersectional factors like age, gender, and culture, which will be discussed in a separate theme, and these can vary across generations [33]. Sexual harassment was not officially recognized until the 1986 case of Meritor Savings Bank v. Vinson; thus, sexual harassment did not gain national public attention in the US until the 1990s (e.g., with the Tailhook scandal and the Clarence Thomas hearings) [33]. The potential for generational differences in awareness of what constitutes power imbalances (e.g., owing to gender) cannot be dismissed. Yet, awareness of power imbalances and how power can be used to coerce, as well as awareness of what constitutes “normal” or acceptable behavior, can influence how SV is understood and subsequently addressed (or not addressed) [2,33].

More recently, Bows (2018) also found that the SV practitioners in her study found it challenging to appreciate the generational social and cultural differences older survivors navigated when they were younger, during a time when public discussions on SV did not occur [1]. This was referenced by service providers in the current study as well, as several service providers noted that SV was not as openly discussed among this current cohort of older adults. One participant described this generation as one in which, “you just didn’t speak up when things like that happened”. Thus, as another service provider pointed out, “social standards” (later described by this participant as ‘social norms’) within given generations can present barriers to preventing SV against people who are 50 and older. As such, awareness and understandings of unacceptable sexual behavior not only vary among researchers but can also vary by geography, culture, and generation, including for professionals who witness SV in later life and/or receive reports of it, which could have influenced the participants’ perspectives on SV against adults 50 and older, particularly given the sample mostly comprising service providers who were younger than 50 years [1,2,33].

In consideration of many older adults growing up “having to keep [quiet]” about SV, more SV resources tailored to older adults were recommended by the service providers. Moreover, a need to learn more about the social norms older adults grew up with was noted, along with further training on the ability to effectively communicate with older adults, to advance prevention. Still, it is noteworthy that the only reference to generational influences on how SV is understood and discussed was identified by service providers who worked in healthcare. Further training on generational contexts and how this may impact reporting may be needed for other professions as well (e.g., for social workers, SV agency workers, and law enforcement).

### 3.2. Needs for Knowledge, Awareness, Research and Education

The participants expressed concerns surrounding persistent needs for awareness and the impacts such needs can have on perceptions of SV, emphasizing that SV, as one participant noted, is “more common than we think”. A need for greater community and organizational awareness (e.g., in long-term care organizations) was identified across industry types. With this awareness, greater awareness of where to report SV once it occurs was recommended, which could aid in preventing further SV from occurring.

Some expressed gratitude for the study as well, noting they did not know SV was a problem in later life prior to the survey, further underscoring needs for more awareness, including among professionals. For example, a participant reported “there are many people… suffering this kind of event. So this study is very helpful to help people realize this issue”. Beyond serving survivors, several participants had survived earlier life SV as well, which may have influenced the gratitude expressed for the ways research can validate lived experiences. One participant who shared they survived SV in later life noted “having someone believe me” was essential, adding “thank you for trying to put something together to help [victims]”.

#### 3.2.1. Influences of Power on Awareness of SV in Later Life

The influence of power on understanding SV was shared as well. Some participants noted that power affords opportunity to frame the narrative, including through mainstream media, at the societal level. One participant stated, “TV has made a huge difference in domestic violence due to victims and aggressors seeing there are relationships modeled without such behavior. Simply being aware… of a different and better way is the first step”.

Power imbalances were particularly noted between men and women, as men were identified as having more power over women to abuse them. Power was also identified as a considerable factor that may influence SV and how it is addressed in work contexts, as imbalances in power were identified as a key barrier to addressing SV. For example, a healthcare worker noted that some victims may be afraid to report SV owing to the fear of losing their job. Still, the possibility to use positions of power to condemn and respond to SV was also mentioned.

#### 3.2.2. Considerations for Education and Awareness

Needs for more education on SV in later life and for greater public awareness were noted as key barriers to preventing SV in later life. A participant shared, “I heard about this happening before when I worked at a nursing home”, in reference to resident-to-resident SV, adding “they had a class about [SV]. But aside from that I’ve never heard it… discussed in a public setting”. Another healthcare service provider simply noted that the public should be better informed about this issue and how to react to it.

In addition, one participant suggested “abuse training should be mandatory for all employees in all organizations”, highlighting needs for awareness at the organizational level. Another recommended “Education! Teaching the people that care for other people that it happens”. This sentiment was common. For example, another participant shared “my only experience with this has been in a nursing home setting. Unfortunately, I have saw several incidents being reported… I have always believed that more classes should be given, and training, on dealing with this”. Not only was a need for further training noted, but the desire to learn more about how to best prevent SV in later life was underscored, as one participant shared, “I work in a care facility so I am interested in the welfare of the elderly people around me. Knowing what to look for… with those elderly people would be incredibly helpful”. Another participant asserted, “Education is needed regarding appropriate vs. inappropriate touching”.

Research was understood as essential to awareness, as one participant noted needs for studies with “accurate statistics” on prevalence, along with “listen[ing] to as many stories as possible to look for patterns” to better understand SV in later life.

Further, a different participant, who worked in law enforcement and shared learning while off duty about a caregiver in a facility sexually abusing an older adult, which he reported to the caregiving agency where the caregiver worked, emphasized the importance of further organizational awareness, noting that some agencies “have no clue this goes on”. This participant later learned that “the caregiver was fired” after the caregiving facility was made aware of the SV occurring. Still, it was not noted whether a police report or APS report was also filed in this case, which may reflect the taboo nature of SV in later life and support the recommendations of other service providers for a clear system for reporting SV that occurs in later life, as other service providers shared that SV in later life is seldom reported.

### 3.3. Needs to Take Sexual Violence Seriously through Policy and Resource Development

Several participants emphasized that SV is not perceived as a serious problem in society, including SV in later life. For example, one participant shared, “people don’t take the reports seriously or really monitor older adults”. Still, another participant stressed, “People still take this lightly and they are victims without knowing”, referencing limited public (or societal-level) knowledge on what constitutes SV, in general, across the lifespan. Limited finances for programs, transportation, and other resources to address SV in later life were highlighted. One participant suggested “better screening of employees who work with older people”. Ongoing needs for more outreach with at-risk communities was also identified as a barrier to prevention along with “more [resources] in all towns and cities”, as one participant noted, “there are not near enough around”. Another added, “Most hospitals and police stations have people trained to deal with sexual abuse. But in small towns like mine there aren’t any places to receive counseling for victims… every town should have a place like that”.

This was attributed to greater surveillance in long-term care facilities and needs for more frequent prosecution. Concerns for improved systems for reporting were also commonly expressed. A participant suggested, “There needs to be more public advertising for ways to report sexual abuse”. A call was made for additional anonymous online reporting options, which may highlight needs for outreach to improve awareness of resources, such as through bystander approaches. Beyond increasing surveillance in nursing homes, which was recommended by several participants, in-home care is in need of further surveillance, enhanced screening and training, and higher-quality care to enhance prevention, as SV in later life not only occurs in nursing homes but it frequently occurs within the community as well [2,18,28]. As such, more resources positioned closer to where older adults live [16] and further efforts to strengthen community connections have been recommended to create more supportive environments [1,16,28].

More supportive work environments were particularly recommended to advance prevention along with SV-related supports for older victims, including through individual and group counseling and peer support [targeted] toward “adults/geriatrics”. For example, a participant suggested that older adults should be afforded “a multitude of support… so they may feel as if they can talk about [SV]”, adding that “people of their age” (in later life) may “help support them” including with “how to talk comfortably about sex”.

### 3.4. Victim Blame and Internalized Stigma

Several participants highlighted that victims are perceived as responsible, as SV is understood as preventable potentially through self-defense or firmer boundaries. One participant underscored that a common perception is that “especially women solicit this through their clothes or behavior”. A participant noted it is widely believed “that somehow, you were asking for it. Whether you were just being nice and polite… that was read as being flirtatious, or whether you were wearing something you look good in… somehow that invites others to touch you”. Yet another shared “a lot of people” believe that victims “must [have] done something to bring the situation upon themselves”. A different participant highlighted that ageism can impact this, as “the victim could be blamed for not being sane or having the right mind or that they are old fashioned”.

A participant shared, “another belief is that no really means yes”, referencing gendered sociocultural influences. Another participant observed that a prevailing perception is “that women deserve it because of the way they dress, that men deserve to have sex whenever they want”, highlighting gendered understandings of SV. Another stressed it is widely believed that “the woman is to blame”, as “men can’t control their sexual desires”. Further, a participant shared, some older adults fear “being made to feel badly, like they should have been more careful”.

Self-internalized victim blame was discussed as well, and in particular, embarrassment was frequently noted. For example, one service provider shared that survivors who are 50 and older “might feel ashamed of what happened or blame themselves for not being more careful” and “too embarrassed to seek out help”. Still, some of the embarrassment older survivors may encounter was attributed by the service providers to “social norms” that promote silence around SV.

Further, the participants called attention to the social taboo surrounding SV in later life and the strong emotions it evokes while expressing anger that SV occurs. The anger and fear participants expressed may be linked with the social taboo surrounding SV in later life. For example, several participants shared that reflecting on SV in later life resulted in discomfort with discussing SV particularly owing to the emotional pain that victims experience. As one participant concluded, “The pain that the women have during that time will make me… cry”. It is possible that multi-level silence, among individuals, organizations, and at the societal level, has resulted from strong emotions and discomfort with the idea that older adults are sexually abused. The potential role of ageism and sexism in this silence and in stigma surrounding SV in later life was noted by several service providers. This reflects research that has highlighted the ways ageism and sexism have influenced the shock that is commonly experienced by service providers [1,5].

The service providers recommended that the taboo surrounding SV in later life must be addressed. One suggestion for addressing this taboo that was offered was the recommendation to encourage more open and more frequent discourse on SV in later life. For example, one participant shared, “when people think about [SV] they usually think about older men and young girls, or young men and… women, but there isn’t much conversation about [SV] in people over 50”, noting more open discussions of SV in later life would result in “people [realizing] they aren’t alone and [speaking] up more”.

### 3.5. Ageism, Intersectional Prejudice and Rape Culture

Ageism was identified as a barrier to preventing SV in later life, as a participant suggested, “older people may not want to address these issues because of embarrassment”. Another participant noted, “The victim could be blamed for not being sane or having the right mind or that they are old fashioned”. It was noted that this is not helped by current stereotypes of elders and sexuality, as one participant concluded, “people believe that older people are not sexual”. Some participants also mentioned that society members generally do not feel comfortable talking with older adults. This can present a substantial barrier to both preventing and addressing SV in later life. Moreover, the participants stressed that older adults must be believed rather than dismissed.

Several participants noted that beyond heightened risks for SV that women experience, older women are vulnerable owing to being older and potentially to experiencing more health issues that are expected with growing older despite SV in later life not being widely acknowledged. As one participant put it, in addition to women being more vulnerable to SV, “often people who are ill, institutionalized, or elderly may be at greater risk due to frailty or chronic health conditions”, suggesting that gender, ability, and age can all increase risks for SV. Accordingly, this same participant cited a need for “more supportive resources” for older adults and highlighted that victims living with dementia are commonly disbelieved, which can impact how SV in later life is addressed.

Another participant described the influence of sexism on women being at a greater risk for SV than men and pointed out that this can be more pronounced depending on “cultural norms” or a victim’s age, adding that people who are older than 50 may be understood as senile and easier to control due to being older, which can result in unique intersectional risks for SV among older adults. This same participant later described learning of SV against an older “bed ridden” woman by a former male colleague in a long-term care facility, highlighting an example of encountering these intersectional risk factors for victimization in her practice with older adults.

A separate participant suggested, “examine attitudes toward death and dying and how they affect the elderly”, adding that “many stereotypes exist surrounding the realities of being an older adult”, thus pointing out the role in age-based stereotypes and potentially disease and dying being prominent factors that can influence how the process of aging is understood. This can present a barrier to interests in learning about issues older adults experience and how to prevent and address them if, for example, younger service providers are afraid of death and dying and associate older adults and growing older with death and dying. Beyond this, other participants observed that limited mobility can raise risks for SV, which may be more pronounced in later life, noting that this is an added risk factor to consider in addition to the greater vulnerability women experience.

Gendered influences on SV were commonly observed across professions, as a participant noted that SV is perceived as a response to “people ask[ing] for it… or that it’s just ‘boys being boys’ or ‘girls being girls’”. Others attributed gendered vulnerability to power and privilege, as a participant stressed, “If men are in power, then these things will never be [acknowledged]”. Another shared, “Powerful men often get away with sexual assault”, noting SV they learned of, after which the survivor remained silent, because the perpetrator “was a powerful man”, adding the victim “has had PTSD since”.

Several participants pointed out that SV perceptions generally involve “men abusing women” although “the roles could be reversed”, as another noted “it is… believed that men [cannot] really experience this kind of abuse”, highlighting that men can be victims despite research on male victims being limited, including in later life.

Culture was also identified as potentially impacting trust or mistrust of authorities, which can limit reporting SV in later life in addition to the other factors that are discussed above. One participant shared “definitely age and the barriers surrounding the topic of sex and sexual assault” impact perceptions of SV, especially in later life. This same participant pointed out that “most younger folk are more open about it than older folk are”, yet this “can also be used for an assaulter’s advantage, as then they are less likely to express or talk about their assault or risk their social standing and image”.

## 4. Discussion

The results highlight that SV is complex and involves violence in general, particularly in later life, as the participants emphasized the taboo surrounding SV, which evokes strong emotions, especially in later life, owing to older adults not being understood as at risk for SV, as the service providers have highlighted. This reflects available research, such as that of Bows (2018) [1], who attributed strong responses to SV in later life among service providers to influences of intersectional ageism and sexism. Consistent with the Critical Feminist Gerontological Framework and with the SEM, several ways age- and gender-based power dynamics can influence perceptions, policies and resources that aid in prevention at individual, relational, organizational and societal levels were identified.

Power imbalances, discrimination, particularly based on age and gender, and stigma can be used to promote external and internalized victim blame, which was identified as a key barrier to preventing SV in later life in particular. The service providers stressed that older adults often feel that no one will believe them (e.g., as one participant put it, that older adults are ‘delusional’, or ‘not in their right mind’ owing to ageist beliefs about older adults being senile). This is supported by the recommendations of researchers as well [1,2,8,16]. Further, this finding supports research that has demonstrated dynamics of power and vulnerability surrounding age, gender and other social demographic characteristics that place individuals at a greater risk for SV [1,16,18], further reflecting the Critical Feminist Gerontological–Social Ecological Perspective that guided this study.

Beyond barriers to reporting, several multi-level perceptions of SV were identified as barriers to prevention (e.g., needs for more outreach and awareness to address age- and gender-based prejudice). Accordingly, further policy and resource development was strongly recommended to improve education and awareness and reduce prejudice. The results suggest the reverse is also true; mis/understandings about SV can impact policies that may aid in prevention and intervention and impede the development, targeted population/s and implementation of prevention and intervention efforts. Thus, if SV is not considered a problem in later life, related training and screening for signs of SV among older healthcare patients or screening for offenders in long-term care facilities or within the community may not be mandated, limiting sexual education and potentially in turn, impacting community health.

### 4.1. Implications for Practice

The findings highlight multi-level stereotypes of SV and older adults, and their influence on how SV is perceived, including in later life, with implications for prevention. At the relational level, older survivors are routinely disbelieved, even by family members, friends, and colleagues. This is supported by extant research [1,2,5,16]. Social workers and other healthcare and social services practitioners can work with community members, family, friends and caregivers to ensure greater support for older survivors and older at-risk populations to enhance prevention. This can be achieved through tailoring bystander interventions for practitioners who work with older adults and being more inclusive of older adults in SV organization trainings. Education and training that challenge personal, organizational and societal stereotypes, namely, through bystander approaches, may also increase knowledge and awareness of where to report SV in later life [1,2].

Results support existing research that suggests SV is a gendered issue across the lifespan and including in later life [18]. Still, participants noted that men can be victims, and further attention is needed to the needs of male and transgender or gender-neutral victims, including in later life. Advocacy is also needed to ensure safe and supportive environments across the life course through challenging stereotypes via community and organizational education. The service providers in this study shared an eagerness to learn how to identify SV in later life, and several service providers noted a desire to learn more about SV in later life, which they had not considered to be a problem in later life prior to the study. Further trainings to educate service providers on how to identify SV against older adults, as the service providers requested, could substantially enhance prevention. Moreover, several participants noted that they did not know how to talk to older adults, and wanted to learn about how to best communicate with older adults, to help address SV when it occurs against adults who are 50 and older. This would in turn prevent further SV from occurring in later life.

In addition, clinical supervisors, administrators and other professionals should be mindful helping clients navigate SV, as it can trigger painful memories among both clients and professionals, and that re-experiencing trauma may impact quality of care. This warrants an ongoing assessment of practitioner needs. It is critical to identify, prevent and address re-traumatization among practitioners (e.g., through vicarious trauma and concerns for workplace safety). Potential benefits from efforts to ensure a safe trauma-informed environment could include greater individual and organizational wellness, productivity and retention, and shifts in societal perceptions over time.

### 4.2. Implications for Policy

Urgent needs were identified by the diverse sample of service providers in this study for community resources, including clear policies to identify risks for, to protect, and to address the needs of older victims. In addition, needs were identified for policies to enhance current screening strategies in order to identify past offenders, including during the hiring process in organizations. In addition, the participants identified needs to prosecute more offenders in order to best prevent and address SV in later life.

Some participants in this study, who worked directly with older adults and with survivors of SV, described shock and disgust when reflecting on the possibility that older adults could be sexually abused, as supported by emerging research [1,2,5]. This supports the recommendations provided in earlier research [1,5] as well as by the service providers in this study that further policies are especially needed to support and promote greater awareness while also addressing ageism and sexism. This may be achieved through requiring organizational trainings on SV that feature a diverse age range of victims and perpetrators and, in turn, challenge stereotypes that older adults are no longer sexual beings and that they cannot be targets of SV.

Outreach efforts are also needed to ensure that survivors and service providers are included in initiatives to prevent and address SV and trauma in later life. This could help with developing and expanding appropriate victim-centered resources. Resource needs were also highlighted for rural areas by the service providers in this study. The participants suggested that hospital workers and law enforcement should be trained to address SV in later life in all small towns in particular.

### 4.3. Implications for Research

While the gratitude shared for the survey was unanticipated, it demonstrates particular value in SV research in later life. Service providers’ comments suggest that conducting research on SV in later life can increase awareness among participants, or in this case, service providers themselves, which has the potential to influence change at the organizational level.

Additionally, the ways SV were understood, in general, across the lifespan, and specifically against adults who are 50+, may have been influenced by generational and sociocultural contexts, which merits further research. For example, in contrast to past social norms surrounding greater public silence on SV, the #MeToo movement has recently begun to address myths about SV by underscoring forms of SV that are not stranger-based, such as IPV-related SV, SV in the workplace, and coercion [2]. This greater awareness of SV has also led to more discussions and, in turn, disclosures of SV [28]. While service providers can offer valuable insight into how SV is understood among helping professionals that diverges from how researchers understand SV, potential generational influences on how SV is understood may have resulted in the service providers describing SV in a way that differs from how at-risk older adults and older survivors may describe SV [1,2,33], which should be further explored.

In terms of the potential impacts of social influences on how service providers understand and respond to SV at the organizational level, it is worth highlighting that while the 2017 #MeToo movement underscored less previously recognized SV contexts, it has focused largely on youth, featuring young, attractive, White, able-bodied female actresses who benefit from multiple forms of privilege that not all SV victims experience [28,34]. Simultaneously, an emphasis has been placed on the vulnerability and victimhood of the young, White, able-bodied actresses [28,35]. Although women are disproportionately victimized and anyone can experience SV, regardless of privilege, this focus, on young, White, able-bodied “perfect” victims was not the intent of the original #MeToo movement, created by Tarana Burke, to help girls and women who live on the margins share their stories as survivors [35]. The recently coopted #MeToo movement has caused harm to survivors and at-risk populations who continue to be marginalized and acknowledged as at-risk. This #MeToo movement (beginning with the hashtag in 2017) did not, for example, advance an awareness of SV in later life but directed further focus on historic “perfect victims” based on ageist, racist, sexist, heterosexist, and ableist social norms [35].

Considering the ways media can influence prevention and intervention across SEM levels, in addition to researching effective training strategies for preventing and addressing SV (e.g., bystander trainings) and testing more inclusive versions of these trainings among service providers, research on other information sources may be warranted. For example, future research may explore the use of social media or other sources for raising awareness of SV in later life among service providers, as one participant recommended drawing attention to SV in later life in “the media”, “PSAs” and in films, underscoring multiple influences on how service providers understand SV. Similar participant recommendations highlight that prevention and intervention research should extend beyond organizational settings, as limited public awareness of SV as a problem in later life can impact how practitioners respond to it and thus prevent further SV in later life from occurring [34].

Longitudinal studies focused on efforts to shift perspectives and advance awareness and prevention are recommended. Providers’ comments demonstrated their view that culture and power can influence whether SV is taboo, influencing victim blame and internalized shame. Further research is also needed on how culture, power and privilege may influence SV perceptions and prevention, especially in later life. Considering the ways different forms of prejudice are linked (e.g., individuals who endorse ageist beliefs are also likely to endorse racist and sexist beliefs) and greater risks for SV among racially non-dominant groups owing to multi-level prejudice, further focus is needed on the prevention and intervention needs of non-dominant populations in particular.

Moreover, the results support the recommendations of Bows (2018) [1] that further interdisciplinary research is needed with practitioners who serve survivors and/or older adults. Service providers in fields of social work, public health, sexual violence, criminal justice, and in work with older adults should collaboratively advance knowledge of the causes and impacts of SV against older adults along with knowledge on how to better support older survivors to encourage recovery and long, healthy lives [1,2].

### 4.4. Limitations

The data for this study were collected online, although several SV researchers prioritize in-person data collection. At the same time, online surveys offer an anonymous means of collecting data on sensitive socially taboo topics that participants may not want to discuss in person. Online surveys also remove scheduling problems for some participants. Yet their multidisciplinary insights are valuable, especially as limited studies exist in this area [1,2].

Participants who selected that they worked with older adults and/or with SV survivors in the survey and who indicated they were healthcare workers, social workers, long-term care workers, administrators or ombudsmen, SV agency workers, and law enforcement officers were included in the survey. While their responses aligned with their reported work type (e.g., participants who indicated they are social workers using person-first language, e.g., ‘adult with disabilities’ as opposed to ‘disabled’ and referencing mandated reporting, and law enforcement officers referring to an ‘assailant’ or ‘suspect’ when discussing the SV in later life they encountered) and they indicated that they worked with older adults or with survivors, there were no other steps taken to ensure that participants had in fact worked with older adults or survivors. Future research may benefit from further consideration of ways to further verify the reported responses of participants.

Another limitation is that data were collected from a mixed group of informants, some of whom had personal experience with SV. As SV in later life is a complex issue in need of further investigation, gathering information from a variety of professional sources, including those with lived experience, provided a broad perspective. However, limiting data collection to service providers from a single discipline may yield more targeted implications that should be considered in future research.

## 5. Conclusions

As the results of this study suggest, SV is a complex social issue that must be further explored and addressed, including in later life. Overarching themes from the participants’ responses surrounded (a) misconceptions of SV in later life and unique barriers to preventing it, (b) needs for further knowledge, awareness, research and education, (c) SV not being taken seriously, warranting further policy and resource development, (d) victim blame and internalized stigma, and (e) ageism, intersectional prejudice, and rape culture. To address the barriers to prevention noted by participants, multi-level prevention measures were recommended, such as individual and group counseling, peer support, organizational policy changes to keep older adults safe, and needs for changes in the ways SV is understood in society, including in later life. Moreover, emphasizing a life-course perspective in sexual education and education on community health needs could advance prevention in this area.

To further understand current needs for preventing SV in later life, more research is needed particularly with service providers who work directly with older adults and survivors. This could promote collaboration among healthcare and social service workers, social workers, law enforcement, and practitioners and administrators from later life and SV arenas. Such efforts have the potential to collectively and comprehensively advance prevention, which as the participants overwhelmingly highlighted, is still especially needed, particularly in later life.

## Data Availability

While the qualitative data collected itself are not publicly available, the analytic strategy and software used for the thematic analysis is publicly available and may be used for replication. This study was not preregistered owing to it being primary research. For further details, please contact the first author at mhand2@gmu.edu.

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
