# Peer review of "Sexual Violence against Adults Aged 50 Years and Older and Implications for Prevention: A Thematic Analysis of Service Providers’ Perceptions"

_ijerph, 2024, doi:10.3390/ijerph21091220_

Round 1

Reviewer 1 Report

Comments and Suggestions for Authors

My overarching thoughts about this paper are that, while it is an interesting topic and idea, the link between the aims and the participants sampled is not clear enough, and the survey used itself is very broad and there are some question marks regarding the wording used. This is reflected in many of the themes where the central aspect of focusing on survivors aged 50 and over is missing. Critically, it’s not made clear why service providers are sampled, and indeed they are asked many questions about their own experiences. These issues play out in the presentation of the results and in the discussion. In addition, the article itself requires a great deal of development and expansion. I do commend the authors for tackling this interesting area of research but at this stage I don’t think this article is ready for publication.

Introduction

And/or older adults – makes it sound as though not all people worked with were survivors? If not, who are these older adults they’ve worked with and how are they relevant to working with older survivors of SV? 

The introduction is very undeveloped and needs to contain far more detail. Sweeping statements such as ‘The age at which later life begins… is important to consider when studying SV in later life’, is a fundamental argument to the paper, but it is not expanded on. Why is it important to consider this section of the population? You’ve noted further down there’s an assumption people aged 50 and older are not sexually abused – where does this come from? There are some interesting points in the introduction but drawing out these arguments in the introduction will strengthen the paper considerably (and see comment below about restructuring).

Your aims require further development to make it clear why service providers are the best people to be asking your research questions to, rather than survivors themselves.

Method 

The section on conceptual framework is detailed and provides some nice information. However, the rest of the method needs to be separated into more detailed sections, participants, procedure etc. Much more detail is required here in line with standard empirical protocols. For instance, it’s not clear how you ensured that your participants have worked with SV survivors aged 50 and over. The focus in the introduction is on healthcare providers but you seem to have included law enforcement officers – this is an important distinction which requires unpacking and justifying as they may have very different views.

In the survey, did you make it clear that you wanted participants to answer all questions with survivors aged 50 and over in mind? At present this is not clear. It would also be helpful to briefly explain the decisions made around wording in the survey e.g. the use of ‘unwanted sexual touch and behavior’. You’ve also asked a lot of questions about the participants as victims themselves – this changes the scope of the survey for me as you’ve previously indicated you’re interested in professionals’ views, rather than victims themselves, and this requires further clarity. It’s also not clear why vignettes were used – expanding on this would be helpful.

Results

The summary of the demographic details is some nice contextual information.

Some of the issues around the lack of focus of the aims or of sample is played out in the results section, which is quite broad and at times unfocused. There is also a lack of information pertaining specifically to survivors aged 50 and over which is a core element of the paper, particularly in relation to what participants may have had to say about this type of SV compared to SV against other ages. Or did they suggest the issues are broadly the same? The section on ageism is the most interesting in this respect.

Discussion

I think you need to be mindful of the way you’re using the word prevention when you’re talking about outreach and awareness, There is a focus on victims here – what about prevention in terms of addressing the offending itself? Equally, does educating professionals prevent SV, or make them more effective at helping survivors once they have already experienced SV? Some expansion and clarification of what prevention could look like would be interesting here.

The discussion would benefit from framing the results in terms of what we know about SV generally and whether the findings show survivors aged 50 years and over to be a distinct group with distinct needs.

Comments on the Quality of English Language

The article could be more clearly written in places, e.g. ‘SV and the trauma’ is not well phrased, and there are multiple instances where rewording would afford the article greater clarity, e.g. first sentence of abstract (and in introduction) – ‘older adults’ – make it clear you mean victims. There are equally some very short paragraphs which require integrating, and the structure of the introduction as a whole would benefit from some reordering so the reader can more easily follow the narrative you’re building. For instance, mentioning your survey details halfway into the introduction is confusing and on first read seems as though you’re outlining previous research.

‘owing to the impacts of trauma and oppression on aging’ – this bit of this sentence is unclear. Is oppression the right word to use? You’ve expanded a bit on this in the Method but without explanation here it seems out of place.

Author Response

Reviewer 1 Overall Comments and Suggestions for Authors:

My overarching thoughts about this paper are that, while it is an interesting topic and idea, the link between the aims and the participants sampled is not clear enough, and the survey used itself is very broad and there are some question marks regarding the wording used. This is reflected in many of the themes where the central aspect of focusing on survivors aged 50 and over is missing. Critically, it’s not made clear why service providers are sampled, and indeed they are asked many questions about their own experiences. These issues play out in the presentation of the results and in the discussion. In addition, the article itself requires a great deal of development and expansion. I do commend the authors for tackling this interesting area of research but at this stage I don’t think this article is ready for publication.

Introductory Response for Reviewer 1: Thank you so much for making time to review this work. We very much appreciate it, as well as your organization of your comments, which have been easy to track when making the recommended revisions. The examples provided in your feedback and your thoughtful level of detail have been very helpful in further expanding on this manuscript and making this manuscript what we now believe is publication-ready.

We agree that it should be clear why service providers were sampled, and why they were asked so many questions for this study, and have now clarified this, beginning on p. 2 (with all revisions highlighted in green in the revised manuscript), explaining:

“This has substantial implications for service providers, who serve older adults as well as for those who serve victims of SV [1, 2, 5; 16]. Service providers can offer valuable information surrounding barriers to prevention and intervention and what is needed to address them [1, 2, 5]. However, very little research exists on the experiences of service providers responding to SV in later life [1, 2]. In terms of where SV in later life is reported, the limited existing research in this area has demonstrated that reports of SV in later life have been made to service providers working in multiple disciplines. This includes health care workers [8], adult protective services (APS) workers and ombudsmen [9], practitioners in long-term care settings [5], social workers [16], service providers in organizations that service SV survivors (e.g., rape crisis centers) [17], and police [16; 18]. The WHO (2016) has referred to each of these diverse service providers (healthcare providers, social service workers, law enforcement, and victim service workers), many of whom regularly witness SV and of whom receive reports of SV, as “key stakehold-ers,” as these service providers are essential to its prevention and intervention, [6]. Across the lifespan, and in especially in later life, further multidisciplinary focus is also needed to prevent and address SV in later life [1, 2, 16].”

Later, when discussing the present study reported in the manuscript, the rationale for surveying the diverse service providers that were included in this study is further explained, on pp. 3-4. Some of these revisions are also highlighted in our response, below.                      

Introduction

  1. Comment 1: And/or older adults – makes it sound as though not all people worked with were survivors? If not, who are these older adults they’ve worked with and how are they relevant to working with older survivors of SV? 

Response 1: Thank you for this question. This is now clarified on p. 4:

“While service providers are essential to prevention and intervention, as for example, their perceptions and knowledge on SV in later life and any needs for further awareness in this area among these professionals can impact service delivery [1, 2, 5], the authors recognize that survivors themselves are key experts on SV in later life. In consideration of this, data was also gathered from survivors and disseminated as part of the first author’s larger dissertation study. In the present manuscript, which is based on the first author’s dissertation study, the findings are presented on the data collected from service providers, who have worked directly with older adults and/or with SV survivors (including healthcare professionals, social service professionals, social workers, SV organization workers, law enforcement, and ombudsmen) owing to these service providers being defined as key stakeholders in preventing and addressing SV [6], including in later life [5, 8, 9, 16, 17, 18].                                                                                          

It is noteworthy that not all service providers that were included in this study exclusively served SV survivors, because research has demonstrated that SV in later life is directly witnessed by healthcare professionals, and social service professionals, and reported to these professionals as well as to SV organization workers, law enforcement, and ombudsmen [5, 8, 9, 16, 17, 18]. Across the lifespan, and in especially in later life, needs for further multidisciplinary focus have also been identified to more comprehensively prevent and address SV in later life [1, 2, 16]. However, as discussed, there is little published research on service provider awareness, knowledge and experiences with learning about SV in later life, and the present study contributed to the limited knowledgebase in this area [1, 2].”              

This is also clarified on p. 2, and the abstract has been edited to emphasize the relevance of including the service providers who generally receive reports of SV in later life, even if they are not working in a SV organization. (The words that were deleted from the abstract to make space for this important distinction are highlighted in blue for easy tracking in the revised manuscript, and for transparency).

  1. Comment 2: The introduction is very undeveloped and needs to contain far more detail. Sweeping statements such as ‘The age at which later life begins… is important to consider when studying SV in later life’, is a fundamental argument to the paper, but it is not expanded on. Why is it important to consider this section of the population? You’ve noted further down there’s an assumption people aged 50 and older are not sexually abused – where does this come from? There are some interesting points in the introduction but drawing out these arguments in the introduction will strengthen the paper considerably (and see comment below about restructuring).

Response 2: Thank you for your feedback. To be transparent, the original manuscript draft had been substantially cut to remain within the recommended manuscript word count while reporting our qualitative results and implications, and most of these cuts were made to the Introduction section prior to submission. The Introduction section has now been expanded upon, for further context, with much of this information added back in. This section has also been reorganized, incorporating your feedback, and we believe that it is much clearer, thanks to this feedback.

The statement on the “the age at which later life begins,” was included owing to gerontologists disagreeing on when later life begins and thus, when researchers should begin studying issues older adults experience, like SV in later life. This is now clarified on pp. 1-2:

“The age at which later life begins, commonly identified as 60 years or older, is important to consider when studying SV in later life, which is often referred to as elder sexual abuse (defined as an unwanted hands-on or hands-off sexual interaction with an older adult) [10]. Gerontologists do not agree on when later life begins or in turn, on when to begin studying SV in later life; thus, the cut-off age in research on SV in later life has varied, as some research has explored the experiences of individuals who are as young as 50 years, owing to findings that suggest non-dominant populations may age faster, due to structural discrimination and related trauma [11, 12, 13]. This is relevant to research on SV in later life, as SV and resulting trauma (a natural yet complex response to an event or events that overwhelm one’s sense of equilibrium due to perceived or actual limited social supports, often through posing threats of serious injury, death, and/or violation of physical integrity) have also been linked with poverty, discrimination, poor health, obesity, and accelerated aging [14]. In other words, aging can depend on various personal experiences (e.g., structural oppression, poverty, and other factors linked with risks for violence) as well as earlier life SV exposure and the physical and mental health conditions that survivors often experience (e.g., poor general health, depression, low-self-esteem, eating disorders, obesity, and obesity-related disorders, like diabetes} [11, 12, 13, 14].              Accordingly, it has been suggested that studying SV among adults 50 and older is likely to be more inclusive of underserved populations when researching SV in later life, because some people may age faster than others [12].                                                                                                                       

As such, this manuscript will focus on adults who are 50+ as the age cut off for defining SV in later life, in adopting the inclusive later life definition provided by The National Clearinghouse on Abuse in Later Life (NCALL), for the reasons discussed above [12]. Further, the focus of this manuscript is on SV in later life in particular, in consideration of it being widely underrecognized, underreported, and understudied, despite SV in later life being a longstanding, growing public health issue [4] with lasting physical and psychological consequences, which service providers (e.g., health care and long-term care workers, social workers, SV service providers, law enforcement, and ombudsmen) can help to prevent and address [5, 6, 8, 9]. The timely relevance of SV in later life to service providers, both those who serve older adults, and those who serve SV survivors, will be further described within this section.”                                

We hope that this and the other tracked revisions in the Introduction section help highlight the consequential focus on this population.

In terms of where the information has come from, on the assumption that older adults are frequently perceived as not sexual, not sexually desirable, and subsequently, not at risk of SV, including by professionals who work with older adults in healthcare, nursing home, and long-term care organizations, Bows, 2018, Iversen et al., 2015, and the previous research of the authors of this study have been cited, along with other scholars in this section and later in the manuscript. We realize that we did not cite these authors in our rationale for our study on p. 2 however, beginning “To address this and the taboo nature of SV past 50 years of age,” and we have now cited these authors here as well, as [1 ,2, 5], for further transparency on where this information has come from. We hope that this is helpful.

  1. Comment 3: Your aims require further development to make it clear why service providers are the best people to be asking your research questions to, rather than survivors themselves.

Response 3: Thank you for directing our attention to this; this is an excellent point. We agree that our stated aims were in need of further clarification. The authors have this clarified on, on p. 4:

“While service providers are essential to prevention and intervention, as for example, their perceptions and knowledge on SV in later life and any needs for further awareness in this area among these professionals can impact service delivery [1, 2, 5], the authors recognize that survivors themselves are key experts on SV in later life. In consideration of this, data was also gathered from survivors and disseminated as part of the first author’s larger dissertation study. In the present manuscript, which is based on the first author’s dissertation study, the findings are presented on the data collected from service providers, who have worked directly with older adults and/or with SV survivors (including healthcare professionals, social service professionals, social workers, SV organization workers, law enforcement, and ombudsmen) owing to these service providers being defined as key stakeholders in preventing and addressing SV [6], including in later life [5, 8, 9, 16, 17, 18].”    

Method 

  1. Comment 4: The section on conceptual framework is detailed and provides some nice information. However, the rest of the method needs to be separated into more detailed sections, participants, procedure etc. Much more detail is required here in line with standard empirical protocols.

Response 4:

  1. Comment 4a: For instance, it’s not clear how you ensured that your participants have worked with SV survivors aged 50 and over.

Response 4a: Thank you for sharing this—we can now see why this would be the case, and have clarified this. We also added added further subheadings to organize this section, which has been expanded upon. And on p. 5, it is now explained:

“Included were participants who selected that they worked with older adults and/or with SV survivors in response to demographic questions on work type and individuals who indicated they were healthcare workers, social workers, long-term care workers, SV agency workers, ombudsmen and law enforcement officers, who are likely to have worked with older adults and SV survivors, owing to the multidisciplinary needs for information on SV past 50 years established in research, and the likelihood of encountering it, considering the nature of SV past 50 years and how reports are made [2].”             

In the Limitations section, it is also now noted:

“Further, participants who selected that they worked with older adults and/or with SV survivors in the survey and who indicated they were healthcare workers, social workers, long-term care workers, SV agency workers, ombudsmen and law enforce-ment officers were included in the survey. While their responses aligned with their reported work type (e.g., participants who indicated they are social workers using person-first language, e.g., ‘adult with disabilities’ as opposed to ‘disabled’ and refer-encing mandated reporting, and law enforcement officers referring to an ‘assailant’ or ‘suspect’ when discussing the SV in later life they encountered) and they indicated that they worked with older adults or with survivors, there were no other steps taken to ensure that participants had in fact worked with older adults or survivors. Future re-search may benefit from further consideration of ways to further verify the reported responses of participants.”

  1. Comment 4b: The focus in the introduction is on healthcare providers but you seem to have included law enforcement officers – this is an important distinction which requires unpacking and justifying as they may have very different views.

Response 4b: Thank you for underscoring this. Near the bottom of p. 6, it is now further explained:

“It is worth noting that while like the other service providers, law enforcement receive reports of SV and are often included in SV research [18], further collaboration with them is recommended for prevention, and both law enforcement healthcare workers have been identified as “key stakeholders” for preventing and addressing SV [6], law enforcement and healthcare workers may have different views on SV in later life. However, aside from the professional language used “e.g., ‘assailant’ instead of ‘per-petrator’ for example), there were no substantial differences found among the different professions with regard to the nature of their responses on their views of SV in later life or what is needed for prevention and intervention, including among the six law en-forcement officers included in the study. As such, the decision was made to include all of the different service providers’ responses in the thematic analysis.”                                                                               

  1. Comment 5: In the survey, did you make it clear that you wanted participants to answer all questions with survivors aged 50 and over in mind? At present this is not clear.

Response 5: On p. 7, it is now noted that a general question about SV asked, followed by more specific questions on SV against people ages 50 and older:

“The survey included general questions about SV (e.g., ‘What (if anything) may people need to know about unwanted sexual touch or unwanted sexual behavior?’) as well as very specific questions about SV against people ages 50 and older (e.g., ‘What (if anything) may be needed to prevent unwanted sexual touch or behavior for people ages 50 and older?’). A list of the survey questions is provided in Appendix A.”

  1. Comment 5a: It would also be helpful to briefly explain the decisions made around wording in the survey e.g. the use of ‘unwanted sexual touch and behavior’.

Response 5a: Thank you for this suggestion. This is now expanded upon, on p. 5, as it is now noted:

“Questions were phrased to be easily understood (e.g., using ‘unwanted sexual touch’ for example, rather than ‘sexual violence,’ with the term ‘unwanted sexual touch’ having been used by other researchers as well, to briefly describe SV in plain language [27]).”

  1. Comment 5b: You’ve also asked a lot of questions about the participants as victims themselves – this changes the scope of the survey for me as you’ve previously indicated you’re interested in professionals’ views, rather than victims themselves, and this requires further clarity.

Response 5b: Thank you for this point. We agree that further clarity was needed. It is now clarified on p. 5:

“The vignettes described SV scenarios involving victims up to 81 years old, as part of a larger dissertation study, to gauge how a victim’s age may impact perceptions of SV (which all participants in the larger dissertation study were asked prior to being asked if they worked with older adults or with survivors and clarifying their occupation). The larger mixed methods dissertation study involved quantitative questions regarding responses to the SV vignettes, to examine how the age of the victim and the type of SV may potentially impact perceived seriousness, culpability, reportability and knowledge of SV. Following this, a thematic analysis was conducted, of the open-text (write-in) survey responses of survivors of later life SV and of service providers who participated larger dissertation survey. Service providers who did not personally experience SV were asked 23 survey questions, and if they had personally experienced SV, they were asked an additional four questions, to explore how (if at all) personal SV experience may have impacted how they understood and responded to SV, as part of the larger dissertation study. (See Appendix A for the survey questions).”

  1. Comment 5c: It’s also not clear why vignettes were used – expanding on this would be helpful.

Response 5c: Thank you for this question. It is now noted on p. 5:

“The vignettes described SV scenarios involving victims up to 81 years old, as part of a larger dissertation study, to gauge how a victim’s age may impact perceptions of SV (which all participants in the larger dissertation study were asked prior to being asked if they worked with older adults or with survivors and clarifying their occupation).”

Results

The summary of the demographic details is some nice contextual information.

  1. Comment 6: Some of the issues around the lack of focus of the aims or of sample is played out in the results section, which is quite broad and at times unfocused.

Response 6:

  1. Comment 6a: There is also a lack of information pertaining specifically to survivors aged 50 and over which is a core element of the paper, particularly in relation to what participants may have had to say about this type of SV compared to SV against other ages. Or did they suggest the issues are broadly the same? The section on ageism is the most interesting in this respect.

Response 6a: Thank you for your feedback. We have substantially revised the results section (with the revisions beginning on p. 9), and have highlighted the unique ways SV against adults who are 50 years and older is uniquely hidden from public discourse. The unique vulnerabilities of SV against people living with dementia are now highlighted as well.

Discussion

  1. Comment 7: I think you need to be mindful of the way you’re using the word prevention when you’re talking about outreach and awareness, There is a focus on victims here – what about prevention in terms of addressing the offending itself? Equally, does educating professionals prevent SV, or make them more effective at helping survivors once they have already experienced SV? Some expansion and clarification of what prevention could look like would be interesting here.

Response 7: Thank you for pointing this out. Prevention in terms of offending itself is now discussed within the Discussion section.

  1. Comment 8: The discussion would benefit from framing the results in terms of what we know about SV generally and whether the findings show survivors aged 50 years and over to be a distinct group with distinct needs.

Response 8: Thank you for this. Further distinctions are now made, between SV across the lifespan, in general, and implications for SV against people who are 50 and older, beginning on p. 13.

Comments on the Quality of English Language

  1. Comment 9: The article could be more clearly written in places, e.g. ‘SV and the trauma’ is not well phrased, and there are multiple instances where rewording would afford the article greater clarity, e.g. first sentence of abstract (and in introduction) – ‘older adults’ – make it clear you mean victims. There are equally some very short paragraphs which require integrating, and the structure of the introduction as a whole would benefit from some reordering so the reader can more easily follow the narrative you’re building. For instance, mentioning your survey details halfway into the introduction is confusing and on first read seems as though you’re outlining previous research.

‘owing to the impacts of trauma and oppression on aging’ – this bit of this sentence is unclear. Is oppression the right word to use? You’ve expanded a bit on this in the Method but without explanation here it seems out of place.

Response 9: Thank you very much for taking the time to review our work and share these incredibly helpful recommendations. We have substantially revised this manuscript and have highlighted all changes in green, incorporating your feedback throughout the revised manuscript. You will find your feedback incorporated in our abstract as well. We hope that this reads much clearer and very much value your time and feedback. It has substantially improved this manuscript.

Reviewer 2 Report

Comments and Suggestions for Authors

One point to address and I believe is significant - In section 3.1 - The complexity of sexual violence. Persons over 50 were raised differently (depending on the country). For example, in the US - Sexual harassment was not a term until 1986. Date rape and acquaintance rape not discussed. Marital rape was still legal in many states until the mid-late 80's. I think section should be expanded.  This is an important study and can help examine trends since the second wave of feminism. However, generational differences have had serious impact on the interpretation of rape and how it is seen by the survivor. 

See - For 1 in 16 US women, their first experience with sexual intercourse was rape, study says | CNN

See - https://www.washingtonpost.com/dc-md-va/2019/09/19/over-million-women-us-say-their-first-sexual-experience-was-rape/

Author Response

Reviewer 2 Comments and Suggestions for Authors

  • (1, Reviewer 2), Comment 1: One point to address and I believe is significant - In section 3.1 - The complexity of sexual violence. Persons over 50 were raised differently (depending on the country). For example, in the US - Sexual harassment was not a term until 1986. Date rape and acquaintance rape not discussed. Marital rape was still legal in many states until the mid-late 80's. I think section should be expanded.  This is an important study and can help examine trends since the second wave of feminism. However, generational differences have had serious impact on the interpretation of rape and how it is seen by the survivor. 

See - For 1 in 16 US women, their first experience with sexual intercourse was rape, study says | CNN

See - https://www.washingtonpost.com/dc-md-va/2019/09/19/over-million-women-us-say-their-first-sexual-experience-was-rape/

Response 1.2: Thank you so much for making time to review this manuscript and for your useful and encouraging feedback, as well as for the resources you provided to help point us in the right direction. We very much agree with your feedback. Section 3.1 (on pp. 8-9) has been expanded to include:

“Still, it is noteworthy that understandings of personal experiences with SV and sexual harassment and perceptions of SV against older clients can be influenced by intersectional factors, like age, gender, and culture, which can vary across generations [32]. Sexual harassment was not even officially recognized until the 1986 case of Meritor Savings Bank v. Vinson, and did not gain national public attention in the US until the 1990s (e.g., with the Tailhook scandal and the Clarence Thomas hearings) [32]. Yet, differences in awareness of power imbalances (e.g., owing to gender) and how power can be used to coerce, as well as awareness of what constitutes “normal” or acceptable behavior can influence how SV is understood and subsequently addressed (or not addressed) [2, 32].                             

More recently, Bows (2018) also found that the SV practitioners in her study found it challenging to appreciate the generational social and cultural differences older survivors navigated when they were younger, during a time when public discussions on SV did not occur [1]. Thus, awareness and understandings of unacceptable sexual behavior not only vary among researchers but can also vary by geography, culture, and generation, including for professionals who witness SV in later life and/or receive reports of it, which could have influenced the participants’ perspectives on SV against adults 50 and older, particularly given the sample mostly being comprised of service providers who were younger than 50 years [1, 2, 32].” 

A continued discussion of this is included in the research implications section on p. 13, summarizing for example:

Additionally, the ways SV were understood, in general, across the lifespan, and specifically against adults who are 50+, may have been influenced by generational and sociocultural contexts, which merits further research. For example, in contrast to past social norms surrounding greater public silence on SV, the #MeToo movement has recently begun to address myths about SV, by underscoring forms of SV that are not stranger-based, such as IPV-related SV, SV in the workplace, and coercion [2]. This greater awareness of SV has also led to more discussions and in turn, disclosures of SV [33]. While service providers can offer valuable insight into how SV is understood among helping professionals that diverges from how researchers understand SV, potential generational influences on how SV is understood may have resulted in the service providers describing SV in a way that differs from how at-risk older adults and older survivors may describe SV [1, 2, 32], which should be further explored.

Thank you—this has substantially strengthened the manuscript.

Reviewer 3 Report

Comments and Suggestions for Authors

Sexual Violence Against Adults Aged 50 Years and Older and Implications for Prevention: A Thematic Analysis of Service Providers’ Perceptions

The topic is very interesting and of great social relevance. As the authors point out, it deserves further research and greater political and institutional concern. Among the worst crimes of sexual violence, violence against children and the elderly are the cruelest forms of abuse. We agree with the authors that it is not a private crime, it is a structural crime that affects all political, legal, institutional, social and political structures in addition to health and care services.

The conceptual framework is correct, the methodology is sufficiently detailed, although in the presentation of results a greater number of exemplifications of the participants' voices would have been appreciated. 

It is possible that more emphasis should have been placed on the political and economic nature of institutions whose mission and function, and their very existence, is to protect and care for the elderly and who lack sufficient resources and personnel trained to do so. Likewise, home care for the elderly does not provide the surveillance and care that it should for the same reasons as in nursing home institutions.

Despite its limits of depth in the treatment of the topic, the research is necessary and deserves to be published.

Author Response

Reviewer 3 Comments and Suggestions for Authors

Sexual Violence Against Adults Aged 50 Years and Older and Implications for Prevention: A Thematic Analysis of Service Providers’ Perceptions

Comment 1.3 (1, Reviewer 3): The topic is very interesting and of great social relevance. As the authors point out, it deserves further research and greater political and institutional concern. Among the worst crimes of sexual violence, violence against children and the elderly are the cruelest forms of abuse. We agree with the authors that it is not a private crime, it is a structural crime that affects all political, legal, institutional, social and political structures in addition to health and care services.

The conceptual framework is correct, the methodology is sufficiently detailed, although in the presentation of results a greater number of exemplifications of the participants' voices would have been appreciated. 

Response 1.3: Thank you so much for making time for reviewing this work and for your strengths-based response. We agree that the manuscript could be strengthened by further highlighting the participants’ voices, and have since further emphasized their voices, starting on p. 8 (with the revisions highlighted in green).

Comment 2: It is possible that more emphasis should have been placed on the political and economic nature of institutions whose mission and function, and their very existence, is to protect and care for the elderly and who lack sufficient resources and personnel trained to do so. Likewise, home care for the elderly does not provide the surveillance and care that it should for the same reasons as in nursing home institutions.

Despite its limits of depth in the treatment of the topic, the research is necessary and deserves to be published.

peer-review-38623589.v2.pdf (Elder Abuse Ontario page)

Response 2.3: Thank you very much for this immensely helpful feedback. These are great points. While we were unable to access the thoughtfully linked pdf, our feedback is now integrated into our revised manuscript. For example, on p. 3, we have now highlighted:

“Societal perceptions have long been acknowledged as contributing risk factors for elder abuse [20]. As the WHO (2002) noted, “cultural norms and traditions… such as ageism, sexism and a culture of violence… are also… recognized as playing an important underlying role” [21] (p. 132). For example, Brozowski and Hall (2010) have pointed out that at the structural level, “global aging is juxtaposed with neoconservative global market forces which encroach on many Western nation welfare policies designed to protect citizens, including social security, health care, and social services for older adults,” who have been referred to “as an expensive burden to the state,” [16], p. 1184. This sentiment was particularly prominent during the COVID-19 pandemic, during which time older adults, especially those who lived on the margins owing to structural oppression, or widespread discrimination, disproportionately lost their lives as a result of ageist beliefs and public health priorities [3]. Such forms of structural discrimination and oppression (referred to as structural ageism when older adults are disproportionately impacted by preventable social problems), can influence the development or lack of resources and human rights (e.g., the right to health and to safety from violence, including from SV in later life) [3, 16, 18]. 

Specifically relating to the structural impacts of public perceptions, older adults are often publicly perceived as incompetent and undesirable; ageist portrayals and youth-focused rape stereotypes may explain why older adults are generally excluded from efforts to prevent and address SV [1, 2, 13]. Beyond impacting policy development, public perceptions of SV in later life may also become internalized by survivors [19]. Social acceptance of SV as a private issue can impede intervention while encouraging silence, stigma, posttraumatic stress, along with limited reporting when SV in later life occurs [20, 2, 3].”   

On p. 10, it is also now noted:

“Beyond increasing surveillance in nursing homes, which was recommended by several participants, in-home care is in need of further surveillance, enhanced screening and training, and higher quality care, to enhance prevention, as SV in later life not only occurs in nursing homes, but it frequently occurs within the community as well [2, 18, 34]. As such, more resources positioned closer to where older adults live [16], and further efforts to strengthen community connections have been recommended, to create more supportive environments [1, 16, 34].”

Thank you. This has certainly strengthened our article.

Reviewer 4 Report

Comments and Suggestions for Authors

Present the participants' demographic characteristics in a table.

What type of qualitative study did you conduct? 

What was the reason for choosing structured questions in a qualitative study?

The selected questions are an attitude questionnaire, and some questions cannot qualitatively assess people's understanding and experience.

How did you determine the sample size?

What were the inclusion and exclusion criteria for your study?

Was the relationship between the sectional identity and the sectional orientation of people measured with SV?

The findings are not consistent with the purpose of the study

Author Response

Reviewer 4 Comments and Suggestions for Authors

  1. Comment 1, 4 (1, Reviewer 4): Present the participants' demographic characteristics in a table.

Response 1: Thank you very much for making time to review this manuscript and for this recommendation. The participants’ demographic characteristics are now presented in Table 2, in Appendix A.

  1. Comment 2.4: What type of qualitative study did you conduct? 

Response 2.4: Thank you for this question. We conducted a thematic analysis of open-text survey data that was collected as part of a larger mixed methods cross-sectional dissertation study. This is now further clarified on pp. 6-7: where we have noted:

“Research has begun to demonstrate the benefits of online qualitative data collection to explore topics that may be difficult to discuss in person, to accommodate multiple schedules, and to eliminate transportation barriers [25, 26]. Online survey data can yield in-depth insights that may not be possible during interviews when studying issues that may be complex or taboo [26]. Thus, researchers are beginning to gather qualitative data online, through MTURK, to explore sensitive topics, such as attitudes toward SV [25], which can be less intimidating than exploring SV during in-person focus group or interviews. In this study, open-text questions within a survey used for a larger dissertation study, were thematically analyzed. The larger dissertation study was a mixed methods study which involved large-scale quantitative analysis, and open text questions were used to garner qualitative data for thematic analysis of service providers’ knowledge, perceptions and experiences with SV in later life. The themes from this qualitative analysis are captured in this manuscript.”

  1. Comment 3.4: What was the reason for choosing structured questions in a qualitative study?

Response 3.4: This is now clarified on p. 7, where we have explained:

While this strategy, of collecting qualitative data through open-text survey questions, has been useful for studying violence against vulnerable populations owing to the sensitive nature of this topic [24], it does limit the questions that could be asked to structured questions. Still, most of the questions were open in nature and participants were asked what else the researchers should know that they had not been asked.”

  1. Comment 4.4: The selected questions are an attitude questionnaire, and some questions cannot qualitatively assess people's understanding and experience.

Response 4.4: Thank you for this feedback. We agree that some questions cannot qualitatively assess understanding and experience, and have highlighted that this study was part of a larger mixed methods survey study and noted on p. 7 for further transparency:

“Still, it is worth mentioning that some questions cannot qualitatively assess people's understanding and experience with SV against adults who are older than 50, as some of the questions pertained to other parts of the larger dissertation study that were not qualitatively analyzed and are not reported in this manuscript [34]. A list of the survey questions is provided in Appendix A.”

  1. Comment 5.4: How did you determine the sample size?

Response 5.4:

Thank you for drawing our attention to the need to clearly discuss this. The sample size was based on the qualitative data and analysis the first author sought to collect for her larger dissertation study, as well as on the minimum number of participants that would be needed to conduct the quantitative analyses for the larger dissertation study, which is now clarified on pp. 6-7:

“Participants were drawn from a convenience sample of MTURK workers in consideration of stigma linked with SV among elders and to minimize potential harm in a low-risk confidential space for offering information on this complex social taboo. The sample size was based not only on the qualitative data and analysis that would be needed to answer the qualitative research questions as part of the larger dissertation study, but also on the minimum number of participants that would be needed to conduct the quantitative analyses for the dissertation study.  

Thus, a statistical approach was used to determine the sample size based on the appropriate number of participants that would to be sampled in order to conduct each quantitative analysis. This was 500 people for the quantitative analysis, with the highest range for qualitative research reaching to as many as 400 or more participants. The 500 participants we aimed to recruit for the quantitative analyses aligned with our qualitative aims as well, as in order to conduct a thematic analysis, 10-50 people are recommended for text that is provided by participants (e.g., through open text survey questions, such as in the survey) [28]. Multiplied by the five kinds of work industry for service providers (healthcare, social work, work in a SV organization, long-term care, or law enforcement), this equaled between 50 and 500 participants for collecting the needed data to establish common patterns that could answer our research questions while not collecting too much data to manage [28].

The service providers whose open-text responses were qualitatively analyzed were a subsample of individuals who participated in the larger mixed methods dissertation survey, who indicated that they were healthcare workers, social workers, long-term care workers, SV agency workers, ombudsmen and law enforcement officers. Further details on the sample are provided within the Results section.”            

  1. Comment 6.4: What were the inclusion and exclusion criteria for your study?

Response 6.4:  Thank you for this question as well. The inclusion and exclusion criteria are now discussed, on p. 7:

“Included were participants who selected that they worked with older adults and/or with SV survivors in response to demographic questions on work type and individuals who indicated they were healthcare workers, social workers, long-term care workers, administrators and ombudsmen, SV agency workers, and law enforcement officers, who are likely to have worked with older adults and SV survivors, owing to the multidisciplinary needs for information on SV past 50 years established in research, and the likelihood of encountering it, considering the nature of SV past 50 years and how reports are made [2].                                                                                                                                                                                           Thus, the sample included healthcare and social service workers who have worked with older adults and/or with survivors of SV. It is worth noting that while like the other service providers, law enforcement receive reports of SV and are often included in SV research [18], further collaboration with them is recommended for prevention, and both law enforcement healthcare workers have been identified as “key stakeholders” for preventing and addressing SV [6], law enforcement and healthcare workers may have different views on SV in later life. However, aside from the professional language used “e.g., ‘assailant’ instead of ‘perpetrator’ for example), there were no substantial differences found among the different professions with regard to the nature of their responses on their views of SV in later life or what is needed for prevention and intervention, including among the six law enforcement officers included in the study. As such, the decision was made to include all of the different service providers’ responses in the thematic analysis.                                                                                                     

Excluded were individuals who answered less than half of the survey questions, listed in Appendix A.      

  1. Comment 7.4: Was the relationship between the sectional identity and the sectional orientation of people measured with SV?

Response 7.4:

Thank you for this question. We are unable to answer this within this manuscript. This is an important question that also appears to be a quantitative question moves beyond the scope of this qualitative study, which was a thematic analysis of service providers’ knowledge, perceptions and knowledge on SV in later life and their experiences with encountering this in their work. We did collect quantitative data as well, as part of the larger dissertation study that this manuscript is based on, which is reported in the dissertation. We have now cited this dissertation study in the manuscript, on p. 8, for readers to refer to for further details. We hope that this is helpful.

  1. Comment 8.4: The findings are not consistent with the purpose of the study

Response 8.4: Thank you for this comment. We have since further clarified our study aims and research questions, on pp. 5-6.

It is clarified that our study aims to explore the knowledge, experiences and perspectives of SV against adults aged 50 and older among service providers who have worked with older adults or victims of SV and/or later life, how age- and gender-based power dynamics my influence how SV past 50 years is understood, possible barriers to prevention, and how they can be addressed. In our thematic analysis, we identified six themes, surrounding the complex nature of SV; needs for knowledge, awareness, research and education; policy and resource development; victim blame and stigma; rape culture and intersectional prejudice, and SV as a taboo linked with strong emotions, and we have further emphasized the recommendations of service providers for enhancing the prevention of SV against adults who are 50 years and older. These service provider recommendations for prevention are further highlighted on pp. 6-11 and on pp. 14-15. We hope this is helpful and very much appreciate your feedback and the ways it has strengthened our manuscript.

Round 2

Reviewer 1 Report

Comments and Suggestions for Authors

In general, this paper is much improved, and I thank the authors for taking the time to make such careful amendments. I think one overarching issue that still remains as part of this paper is that it seems as though there are several layers being measured, that either need to be simplified when outlining the study, or that need to be drawn out further in the analyses. For instance, I still think that there are claims of several layers of participants’ views being measured here that, if you’re going to mention them as being important, need unpicking. I.e. you have a) service providers who work with older people, b) service providers who work specifically with older SV survivors, and c) service providers who are also survivors. Through the introduction there are some intimation that these different groups may hold different views, and yet this isn’t unpicked in the analysis. If this isn’t going to be addressed, they can’t really be outlined as potentially different in the introduction. Equally, you’ve got several research questions relating to people’s knowledge, experience, and perceptions of SV, as well as barriers to prevention and addressing SV and how they should be addressed, although these are only discussed at a general level in the results (I think this is fine, unless you’re claiming that this is an integral part of the results). Finally, you mention in several places the focus on power dynamics and how these impact on knowledge and perceptions, but this is also not really covered in any depth (you have started to and what you have is very interesting, and you have provided some interesting commentary in the Discussion – but given the focus placed on it in the introduction and aims this could be substantially expanded). In essence, the results you have are interesting, but they almost seem a little underwhelming given the claims that are being made. Simplifying these would allow the reader to see the results as interesting in their own right, without so much complication.

INTRODUCTION

The section on defining 'later life' is interesting but would benefit from being a paragraph in its own right.

The section in green added on the bottom of P2 – I think I understand what you’re saying, but it’s a little unclear whether you mean the lack of research prevents service workers from doing their jobs, or whether service providers are a resource we could tap into to increase our research into and understanding of SV. How does a lack of understanding prevalence affect service providers – might they not be the people that have a better understanding of prevalence?

It's good to see some additional rationale for the study and this section is improved – the additional context that’s been added in is needed. The structure of the introduction is improved but could still follow a stronger narrative structure – perhaps using subheadings may assist with this?

METHOD

Subheadings make this section much easier to read, and there is much more clarity on the sample. Out of interest, are you able to tell the reader how many people, in practice, either work specifically with SV survivors as a core part of their job, or as part of their job, or how many have never worked with SV survivors and so aren’t answering from a place of direct experience?

A couple of points here:

-          The purpose and research questions should be at the end of the introduction – moving this there may result in some repetition that could be streamlined.

-          You’ve clarified the questions asked are both generally about SV and about SV against older people. For the purposes of this study, were only the latter analysed?

-          Equally, it’s now clearer who was asked what questions, but are the questions where people are answering as survivors, not at service providers, relevant here? You’ve noted ‘as some of the questions pertained to other parts of the larger dissertation study that were not qualitatively analyzed and are not reported in this manuscript’ so I assume some questions have been disregarded, although this sentence feels like this is a slightly overcomplicated way of saying ‘Only those questions that related specifically to people’s views on survivors aged 50 or over were included’, and then make sure the questions are highlighted in some way in the Appendix.

No substantial differences found between different professions – are you able to expand a little on how this was established?

RESULTS

The result section has been improved substantially and I thank the authors for the additional information they have added which is very interesting context. Now that the results have been expanded, some consideration could be given to the themes. ‘The complex nature of sexual violence’, for instance, covers a misconception around prevalence, medical issues that may lead to credibility issues, and increased vulnerability. These are all really interesting points and may merit being considered in their own right (and perhaps some of the other information could be streamlined, e.g. it’s not clear what the paragraphs starting ‘The participants also shared that members of the general public’ and ‘Reflections on past SV were also common’ bring to the Results section).

You’ve noted that SV against older clients can be influenced by intersectional factors. How? And what did participants say specifically about this? E.g. talking about power imbalances due to gender infers that women may be more susceptible to SV, but this isn’t expressly said or evidenced by participants’ views. One participant mentioned ‘social standards’ – can you provide more context on what was meant here? There is some information on power dynamics buried in the ‘Needs for Knowledge, Awareness, Research and Education’ theme – again, can this be drawn out separately?

The sections on myths and on rape culture – there are elements of how survivors’ ages are directly related to these concepts but the authors could lead with them in these sections, rather than leading with general information. You’ve also got a theme on SV as a ‘taboo’ – how is this different from SV as a stigma? Perhaps these could be combined?

DISCUSSION

There have been some relevant and well written additions to the discussion.

Some smaller points:

-          How are adults who are at risk of SV defined?

-          P2 – are service providers preventing lasting consequences of SV? Or helping to mitigate their severity / treat issues?

-          P2 – space needed between to and 17%

-          P7 – ‘While this strategy, of collecting qualitative data through open-text survey questions, has been useful…’ – remove commas.

-          P9 – separate theme, not them

-          Last two sentences on P9 needs combining / reworking.

-          P11 – top of the page – did the law enforcement participant report the SV, or did they handle the report?

-          P15 – researching training, not trainings

Author Response

Response to Review 1:

(Thank you so much for dedicating more time to reviewing this work—we so appreciate it)

General Comments: In general, this paper is much improved, and I thank the authors for taking the time to make such careful amendments. I think one overarching issue that still remains as part of this paper is that it seems as though there are several layers being measured, that either need to be simplified when outlining the study, or that need to be drawn out further in the analyses. For instance, I still think that there are claims of several layers of participants’ views being measured here that, if you’re going to mention them as being important, need unpicking. I.e. you have a) service providers who work with older people, b) service providers who work specifically with older SV survivors, and c) service providers who are also survivors. Through the introduction there are some intimation that these different groups may hold different views, and yet this isn’t unpicked in the analysis. If this isn’t going to be addressed, they can’t really be outlined as potentially different in the introduction. Equally, you’ve got several research questions relating to people’s knowledge, experience, and perceptions of SV, as well as barriers to prevention and addressing SV and how they should be addressed, although these are only discussed at a general level in the results (I think this is fine, unless you’re claiming that this is an integral part of the results). Finally, you mention in several places the focus on power dynamics and how these impact on knowledge and perceptions, but this is also not really covered in any depth (you have started to and what you have is very interesting, and you have provided some interesting commentary in the Discussion – but given the focus placed on it in the introduction and aims this could be substantially expanded). In essence, the results you have are interesting, but they almost seem a little underwhelming given the claims that are being made. Simplifying these would allow the reader to see the results as interesting in their own right, without so much complication.

Overarching Response to General Comments:

Thank you so much for making time to review our manuscript again. We very much appreciate it, and understand how important your dedication to excellence is, particularly having recently recommended a round of second revisions for a paper as a peer reviewer that was still in need of improvement. Your recommendations have certainly made our manuscript much stronger, and the results are now much more clearly presented in a more organized way in particular.  All revisions are tracked, with this second round of revisions highlighted in yellow, and additional deleted content highlighted in grey, for clear tracking.

Of note, it is now clarified that “it is possible” that law enforcement officers may have different views from other service providers about SV in later life (on p. 7), and similarities in recommendations across service provider industry types are now discussed, starting on p. 11. In doing so, the unique contributions of some industry types are highlighted, as for example, it is now noted that health care providers were the only service providers to discuss the ways dementia may impact prevention. Power imbalances are further reflected on as well (e.g. on, p. 12). And most importantly, the Results section has been substantially revised, incorporating all of your feedback, which has considerably strengthened this section. All of your feedback was integrated throughout the paper as well. As a result of these improvements, we believe this is a much stronger and better organized paper, and are excited to share this stronger, more polished manuscript. Thank you for helping us with making these considerable improvements.

INTRODUCTION

Comment 1: The section on defining 'later life' is interesting but would benefit from being a paragraph in its own right.

Response 1: Thank you for pointing this out. On p. 2, this section is highlighted as a paragraph in its own right, which is now under the sub-heading ‘Defining Later Life.’

Comment 2: The section in green added on the bottom of P2 – I think I understand what you’re saying, but it’s a little unclear whether you mean the lack of research prevents service workers from doing their jobs, or whether service providers are a resource we could tap into to increase our research into and understanding of SV. How does a lack of understanding prevalence affect service providers – might they not be the people that have a better understanding of prevalence?

Response 2: Thank you for sharing that this is unclear. This section has been revised for further clarity (on p. 3), to now read:

“very little research exists on the experiences of service providers who respond to SV in later life, or on how their practice wisdom may be used to help advance prevention and intervention [1, 2]. Service providers can help enhance our understanding of SV as researchers, which can result in improved policies to prevent and address SV in later life [2, 6]. A more accurate understanding of SV in later life could be used to inform research on prevalence, and service providers themselves can help provide more precise prevalence estimates in consideration of roughly 30% of the SV cases in later life remaining unreported and service providers learning of unreported SV [2, 11]. Additionally, a more accurate understanding of SV in later life (e.g., gained through including service providers from multiple disciplines in research on SV in later life) could influence practice and the trainings practitioners receive [2]. A more well-rounded understanding of SV in later life could also be used to work toward a transdisciplinary understanding of SV in later life, based on service provider knowledge and wisdom [2, 6].”

Comment 3: It's good to see some additional rationale for the study and this section is improved – the additional context that’s been added in is needed. The structure of the introduction is improved but could still follow a stronger narrative structure – perhaps using subheadings may assist with this?

Response 3: Thank you for this feedback and suggestion. Subheadings have how been added (throughout pp. 2-4) We believe this has improved the organization of this section, and appreciate this recommendation.

METHOD

Comment 4: Subheadings make this section much easier to read, and there is much more clarity on the sample. Out of interest, are you able to tell the reader how many people, in practice, either work specifically with SV survivors as a core part of their job, or as part of their job, or how many have never worked with SV survivors and so aren’t answering from a place of direct experience?

Response 4:

Thank you for this feedback. On page 9, it is also now noted:

Of these 126 service providers, 26 (21%) indicated that they worked once or more per week with SV survivors, 34 (27%) indicated that they worked once or more per month with SV survivors, 32 (25%) indicated that they worked once or more per year with survivors, and 34 (27%) indicated that they did not work with SV survivors at all in their field of work, although they did worked in fields that served older adults (who could be victimized). So, the majority of the service providers (73%) indicated that they did work with SV survivors in some capacity as part of their work, although 27% were not answering the questions based on direct practice experience with survivors, but rather in the context of working with older adults who may be at risk of SV.

Comment 5.1: A couple of points here:

-          The purpose and research questions should be at the end of the introduction – moving this there may result in some repetition that could be streamlined.

Response 5.1: Thank you for suggesting this—it reads much cleaner after moving the purpose and research questions to pp. 5-6, at the end of the Introduction section.

Comment 5.2: You’ve clarified the questions asked are both generally about SV and about SV against older people. For the purposes of this study, were only the latter analysed?

Response 5.2: Thank you for your continued dedication to ensuring that this manuscript shines. It has made me a better writer in the process, which is very much appreciated. I can see the need for this clarification now, and have added (on p. 7):

“For the larger dissertation study which this manuscript is based upon, questions were asked are both generally about SV and about SV against people who are 50+ years of age. For the purposes of this manuscript, the analysis of the latter questions, focused on SV against people who are 50+ years of age, are reported.”

Comment 5.3: Equally, it’s now clearer who was asked what questions, but are the questions where people are answering as survivors, not at service providers, relevant here? You’ve noted ‘as some of the questions pertained to other parts of the larger dissertation study that were not qualitatively analyzed and are not reported in this manuscript’ so I assume some questions have been disregarded, although this sentence feels like this is a slightly overcomplicated way of saying ‘Only those questions that related specifically to people’s views on survivors aged 50 or over were included’, and then make sure the questions are highlighted in some way in the Appendix.

Response 5.3:, On p. 8, it is now clarified that “Only those questions that related specifically to people’s views on survivors aged 50 or over were included in the analysis that is reported in this manuscript. A list of the survey questions is provided in Appendix A, in with an asterisk to the left of those questions pertaining to the analysis that is reported in this manuscript, focused on service provider knowledge on SV in later life, barriers to prevention, and recommendations for preventing SV against adults older than 50 years.” Then, in Appendix A, asterisks are used to denote these questions. Thank you for this suggestion—this makes sense, and we agree that transparency is very important.

Comment 5.4: No substantial differences found between different professions – are you able to expand a little on how this was established?

Response 5.4: On p. 7, it is now clarified: “This was determined by closely reviewing the responses of participants from each profession, organized by profession in Microsoft Excel, to explore potential differences between responses. No substantial differences were found regarding how they defined SV in later life, barriers to prevention, or what they believed was needed to prevent it.”

RESULTS

Comment 6.1: The result section has been improved substantially and I thank the authors for the additional information they have added which is very interesting context. Now that the results have been expanded, some consideration could be given to the themes. ‘The complex nature of sexual violence’, for instance, covers a misconception around prevalence, medical issues that may lead to credibility issues, and increased vulnerability. These are all really interesting points and may merit being considered in their own right (and perhaps some of the other information could be streamlined, e.g. it’s not clear what the paragraphs starting ‘The participants also shared that members of the general public’ and ‘Reflections on past SV were also common’ bring to the Results section).

Response 6.1: Thank you very much for your feedback. Upon further reviewing the data, it is clear that that data clearly reflects sub-themes of medical issues and credibility, vulnerability, and generational influences on how SV is understood. Your recommendation to clearly identify and highlight these subthemes is so appreciated and is now reflected on pp. 10-11, where they each now have their own sub-headings as well. We believe this makes the Results section much stronger.

We also agree that the perceptions of members of the general public and the service providers reflecting on their past personal experiences with sexual violence are not noteworthy for this manuscript and have deleted this (from p. 11), incorporating your (immensely helpful) feedback.

Comment 6.2: You’ve noted that SV against older clients can be influenced by intersectional factors. How? And what did participants say specifically about this?

Response 6.2: On p. 15, intersectional factors are further elaborated on, with further quotes from participants and summaries of their discussions of mobility issues and the belief that older adults are senile potentially raising risks for SV in later life in addition to the greater vulnerability women already experience. Thank you for asking about this—we agree that this further discussion was needed.

Comment 6.3: E.g. talking about power imbalances due to gender infers that women may be more susceptible to SV, but this isn’t expressly said or evidenced by participants’ views. One participant mentioned ‘social standards’ – can you provide more context on what was meant here? There is some information on power dynamics buried in the ‘Needs for Knowledge, Awareness, Research and Education’ theme – again, can this be drawn out separately?

Response 6.3: The gender-based power imbalances and greater risks for women are now highlighted in the results, citing the participants’ views (e.g., at the bottom of p. 12 and later on p. 15, when ageism, intersectional prejudice and rape culture is discussed, as this was also relevant to intersectional prejudice). It is also now clarified that the participant who noted “social standards” meant generational social norms (e.g., of not talking about SV), and further elaboration is provided on this, on p. 12. On p. 12, a sub-heading is also provided, on Influences of Power on Awareness of SV in Later Life, to highlight how power can influence awareness of SV in later life and how it is addressed.

Comment 6.4: The sections on myths and on rape culture – there are elements of how survivors’ ages are directly related to these concepts but the authors could lead with them in these sections, rather than leading with general information.

Response 6.4:  We completely agree—thank you for pointing this out. This section is now revised, and ageism is emphasized earlier as well, on p. 15.

Comment 6.5: You’ve also got a theme on SV as a ‘taboo’ – how is this different from SV as a stigma? Perhaps these could be combined?

 Response 6.5: Distinctions were initially made between taboo and stigma, in consideration of the nature of SV in later life being socially taboo and not discussed, while stigma was highlighted to explain responses to SV that were described, which were often noted as resulting in feelings of shame. So, we identified this theme to explain both the way survivors are treated and how they internalize SV. Upon reflection on your feedback and closely reviewing the data again and the rationale for separating the two, we were reminded that the internalization has been a key aspect of stigma in these cases. So, for clarity, we rephrased this, as “Victim Blame and Internalized Stigma” (on p. 13), and we can now see how a discussion of the taboo nature of SV in later life can also fit under this. So, we combined this discussion (which is now on pp. 14-15), to better organize the Results section, in incorporating your helpful recommendations. Thanks to your recommendations, which prompted these changes, we believe the Results discussion is much stronger and much better organized. We can now see why this was needed. Thank you very much for this.

DISCUSSION

There have been some relevant and well written additions to the discussion.

Some smaller points:

Comment 7.1: How are adults who are at risk of SV defined?

Response 7.1:  Thank you so much for your thoughtful and encouraging feedback. On p. 1, it is now clarified that all older adults are at risk for SV, and that prejudice (e.g., sexism) places certain older adults (e.g., women and transgender individuals) at disproportionate risks for SV, including in later life. Risks related to experiencing SV earlier in life and risks related to dementia are also now introduced here.

Comment 7.2: P2 – are service providers preventing lasting consequences of SV? Or helping to mitigate their severity / treat issues?

Response 7.2:  Thank you for pointing out the need for further clarity surrounding this. On p. 2, it is now noted that “Service providers can help achieve this by preventing SV from occurring in later life, which can in turn prevent the sexual trauma and lasting consequences that result from it, and service providers can also treat the lasting consequences of SV in later life.”

Comment 7.3: P2 – space needed between to and 17%

Response 7.3:  Thank you. A space is now provided between them, on p. 2.

Comment 7.4: P7 – ‘While this strategy, of collecting qualitative data through open-text survey questions, has been useful…’ – remove commas.

Response 7.4:  The commas are now removed, with the tracked removal of them (highlighted in grey on p. 8) for easy tracking.

Comment 7.5: P9 – separate theme, not them

Response 7.5:  Thank you for this. This is now updated to read “separate theme” on p. 11.

Comment 7.6: Last two sentences on P9 needs combining / reworking.

Response 7.6:  We agree after reviewing your feedback and the sentences, and have re-worked these sentences on the potential for generational differences in awareness of power imbalances that may have been influenced by policy, on p. 11.

Comment 7.7: P11 – top of the page – did the law enforcement participant report the SV, or did they handle the report?

Response 7.7:  Thank you for asking about this. In this case, the law enforcement participant actually made the report rather than handling it. For further transparency and clarity, it is now noted on p. 13 that this law enforcement participant learned about a caregiver in a facility sexually abusing an older adult, while he was off duty, so he reported the SV to the caregiving agency where the caregiver worked. He did not note whether he reported this to another law enforcement officer or APS worker however, which is also now discussed.

Comment 7.8: P15 – researching training, not trainings

Response 7.8:  Thank you. This is now re-worded on p. 18 as “researching effective training strategies.” Thank you very much for your time and commitment to making this manuscript shine. This is a topic that we are very passionate about, and we so appreciate this, as we want to get this right and believe this manuscript is now in much better shape.